# Identification of DAXX as a restriction factor of SARS-CoV-2 through a CRISPR/Cas9 screen

Alice Mac Kain [1,12], Ghizlane Maarifi[2,12], Sophie-Marie Aicher[3], Nathalie Arhel [2], Artem Baidaliuk [4], Sandie Munier[5,6], Flora Donati [5,6], Thomas Vallet [1], Quang Dinh Tran [1], Alexandra Hardy [1], Maxime Chazal [3], Françoise Porrot [7], Molly OhAinle [8], Jared Carlson-Stevermer [9], Jennifer Oki[9], Kevin Holden [9], Gert Zimmer [10], Etienne Simon-Lorière [4], Timothée Bruel [7], Olivier Schwartz [7], Sylvie van der Werf [5,6], Nolwenn Jouvenet [3✉], Sébastien Nisole [2✉], Marco Vignuzzi [1✉] & Ferdinand Roesch [1,11✉]

Interferon restricts SARS-CoV-2 replication in cell culture, but only a handful of Interferon Stimulated Genes with antiviral activity against SARS-CoV-2 have been identified. Here, we describe a functional CRISPR/Cas9 screen aiming at identifying SARS-CoV-2 restriction factors. We identify DAXX, a scaffold protein residing in PML nuclear bodies known to limit the replication of DNA viruses and retroviruses, as a potent inhibitor of SARS-CoV-2 and SARS-CoV replication in human cells. Basal expression of DAXX is sufficient to limit the replication of SARS-CoV-2, and DAXX over-expression further restricts infection. DAXX restricts an early, post-entry step of the SARS-CoV-2 life cycle. DAXX-mediated restriction of SARS-CoV-2 is independent of the SUMOylation pathway but dependent on its D/E domain, also necessary for its protein-folding activity. SARS-CoV-2 infection triggers the re-localization of DAXX to cytoplasmic sites and promotes its degradation. Mechanistically, this process is mediated by the viral papain-like protease (PLpro) and the proteasome. Together, these results demonstrate that DAXX restricts SARS-CoV-2, which in turn has evolved a mechanism to counteract its action.

---

[1] Institut Pasteur, Université de Paris Cité, CNRS UMR 3569, Viral populations and pathogenesis Unit, F-75015 Paris, France. [2] Institut de Recherche en Infectiologie de Montpellier (IRIM), , Université de Montpellier, CNRS, 34090 Montpellier, France. [3] Institut Pasteur, Université de Paris Cité, CNRS UMR 3569, Virus sensing and signaling Unit, F-75015 Paris, France. [4] Institut Pasteur, G5 Evolutionary genomics of RNA viruses, F-75015 Paris, France. [5] Institut Pasteur, Université de Paris Cité, CNRS UMR 3569, Molecular Genetics of RNA Viruses Unit, F-75015 Paris, France. [6] Institut Pasteur, CNR Virus des infections respiratoires, F-75015 Paris, France. [7] Institut Pasteur, Université de Paris Cité, CNRS UMR 3569, Virus and Immunity, F-75015 Paris, France. [8] Divisions of Human Biology, Fred Hutchinson Cancer Research Center, Seattle, WA, USA. [9] Synthego Corporation, 3565 Haven Avenue, Menlo Park, CA 94025, USA. [10] Institute of Virology and Immunology, Bern & Mittelhäusern, Switzerland, and Department of Infectious Diseases and Pathobiology, Vetsuisse Faculty, University of Bern, Bern, Switzerland. [11] UMR 1282 ISP, INRAE Centre Val de Loire, Nouzilly, France. [12] These authors contributed equally: Alice Mac Kain, Ghizlane Maarifi. ✉email: nolwenn.jouvenet@pasteur.fr; sebastien.nisole@inserm.fr; marco.vignuzzi@pasteur.fr; ferdinand.roesch@inrae.fr

Severe Acute Respiratory Syndrome Coronavirus 2 (SARS-CoV-2) is the causative agent of COVID-19 and the third coronavirus to cause severe disease in humans after the emergence of SARS-CoV in 2002 and Middle East Respiratory Syndrome-related Coronavirus (MERS-CoV) in 2012. Since the beginning of the pandemic, SARS-CoV-2 has infected more than 500 million people and claimed at least 6 million lives. While the majority of infected individuals experience mild (or no) symptoms, severe forms of COVID-19 are associated with respiratory failure, shock and pneumonia. Innate immune responses play a key role in COVID-19 pathogenesis: immune exhaustion[1] and reduced levels of type-I and type-III interferons (IFN) have been observed in the plasma of severe COVID-19 patients[2,3]. Imbalanced immune responses to SARS-CoV-2, with a low and delayed IFN response coupled to early and elevated levels of inflammation, have been proposed to be a major driver of COVID-19[4,5]. Neutralizing auto-antibodies against type-I IFN[6] and genetic alterations in several IFN pathway genes[7] have also been detected in critically ill COVID-19 patients. These studies highlight the crucial need to characterize the molecular mechanisms by which IFN effectors may succeed, or fail, to control SARS-CoV-2 infection.

Although SARS-CoV-2 has been described to antagonize the IFN pathway by different mechanisms involving the viral proteins ORF3b, ORF9b ORF6, and Nsp15[8], detection of SARS-CoV-2 by the innate immune sensor MDA5[9,10] leads to the synthesis of IFN and expression of IFN Stimulated Genes (ISGs) in human airway epithelial cells[4]. IFN strongly inhibits SARS-CoV-2 replication when added in cell culture prior to infection[11,12] or when administered intranasally in hamsters[13], suggesting that some ISGs might have antiviral activity[14]. Relatively few ISGs with antiviral activity against SARS-CoV-2, however, have been identified so far. For instance, spike-mediated viral entry and fusion is restricted by LY6E[15,16] and IFITMs[17,18]. Mucins have also been suggested to restrict viral entry[19]. ZAP, which targets CpG dinucleotides in RNA viruses, also restricts SARS-CoV-2, albeit moderately[20]. OAS1 has been recently identified in an ISG overexpression screen to restrict SARS-CoV-2 replication, through the action of RNAseL, both in cell lines and in patients[21]. Another overexpression screen identified 65 ISGs as potential inhibitors of SARS-CoV-2[22], and found that BST-2 is able to restrict viral budding, although this activity is counteracted by the viral protein ORF7a. We hypothesize that additional ISGs with antiviral activity against SARS-CoV-2 remain to be discovered. Other antiviral factors that are not induced by IFN may also inhibit SARS-CoV-2: for instance, the RNA helicase DDX42 restricts several RNA viruses, including SARS-CoV-2[23]. While several whole-genome CRISPR/Cas9 screens identified host factors required for SARS-CoV-2 replication[24–29], none focused on antiviral genes.

In this work, we performed a CRISPR/Cas9 screen designed to identify restriction factors for SARS-CoV-2, assessing the ability of 1905 ISGs to modulate SARS-CoV-2 replication in human epithelial lung cells. We report that the Death domain-associated protein 6 (DAXX), a scaffold protein residing in PML nuclear bodies[30] and restricting DNA viruses[31] as well as retroviruses[32,33], is a potent inhibitor of SARS-CoV-2 replication. SARS-CoV-2 restriction by DAXX is largely independent of the action of IFN, and unlike most of its other known activities, of the SUMOylation pathway. Within hours of infection, DAXX relocalizes to sites of viral replication in the cytoplasm, targeting an early, post-entry step of the viral life cycle such as viral transcription or uncoating. We show that the SARS-CoV-2 papain-like protease (PLpro) induces the proteasomal degradation of DAXX, demonstrating that SARS-CoV-2 developed a mechanism to evade, at least partially, the restriction imposed by DAXX.

**Table 1 Gene editing efficiency.**

| Gene | % of alleles edited |
|---|---|
| LY6E | 96 ± 1.73 |
| DAXX | 79,67 ± 2.52 |
| APOL6 | 99 ± 0 |
| HERC5 | 97 ± 0 |
| CTSL | 91 ± 1 |
| IFI6 | 88,33 ± 0.58 |
| IFNAR1 | 76,67 ± 3.21 |

The frequency of editing was determined using Sanger sequencing and ICE analysis. Values are represented as mean ± SD (n = 3).

## Results

**A restriction factor-focused CRISPR/Cas9 screen identifies genes potentially involved in SARS-CoV-2 inhibition.** To identify restriction factors limiting SARS-CoV-2 replication, we generated a pool of A549-ACE2 cells knocked-out (KO) for 1905 potential ISGs, using the sgRNA library we previously developed to screen HIV-1 restriction factors[34]. This library includes more ISGs than most published libraries, as the inclusion criteria was less stringent (fold-change in gene expression in THP1 cells, primary CD4 + T cells or PBMCs ≥ 2). Therefore, some genes present in the library may not be ISGs per se in A549 cells. Transduced cells were selected by puromycin treatment, treated with IFNα and infected with SARS-CoV-2. Infected cells were immuno-labelled with a spike (S)-specific antibody and analyzed by flow cytometry. As expected[11,12], IFNα inhibited infection by sevenfold (Fig. S1). Infected cells were sorted based on S expression (Fig. 1a), and DNA was extracted from infected and non-infected control cells. Integrated sgRNA sequences in each cell fraction were amplified by PCR and sequenced by Next Generation Sequencing (NGS). Statistical analyses using the MAGeCK package[35] led to the identification of sgRNAs significantly enriched or depleted in infected cells representing antiviral and proviral genes, respectively (Fig. 1b). Although our screen was not designed to study proviral genes, we did successfully identify the well-described SARS-CoV-2 co-factor cathepsin L (CTSL)[36], validating our approach. USP18, a gene encoding a negative regulator of the IFN signaling pathway[37], and ISG15, which favors Hepatitis C Virus replication[38], were also identified as proviral ISGs. Core IFN pathway genes such as those encoding for the IFN receptor (IFNAR1), STAT1, and STAT2, were detected as antiviral factors, further validating our screening strategy. LY6E, a gene previously described to encode an inhibitor of SARS-CoV-2 entry[15,16], was also a significant hit. Moreover, our screen identified APOL6, IFI6, DAXX and HERC5, genes that are known to encode proteins with antiviral activity against other viruses[39–42], but had not previously been studied in the context of SARS-CoV-2 infection. For all these genes except APOL6, individual sgRNAs were consistently enriched (for antiviral factors) or depleted (for proviral factors) in the sorted population of infected cells, while non-targeting sgRNAs were not (Fig. 1c).

**LY6E and DAXX display antiviral activity against SARS-CoV-2.** To validate the ability of the identified hits to modulate SARS-CoV-2 replication in human cells, we generated pools of A549-ACE2 knocked-out (KO) cells for different genes of interest by electroporating a mix of 3 sgRNA/Cas9 ribonucleoprotein (RNP) complexes per gene target. Levels of gene editing were above 80% in all of the A549-ACE2 KO cell lines, as assessed by sequencing of the edited loci (Table 1). As controls, we used cells KO for IFNAR1, for the proviral gene CTSL or for the antiviral gene LY6E, as well as cells

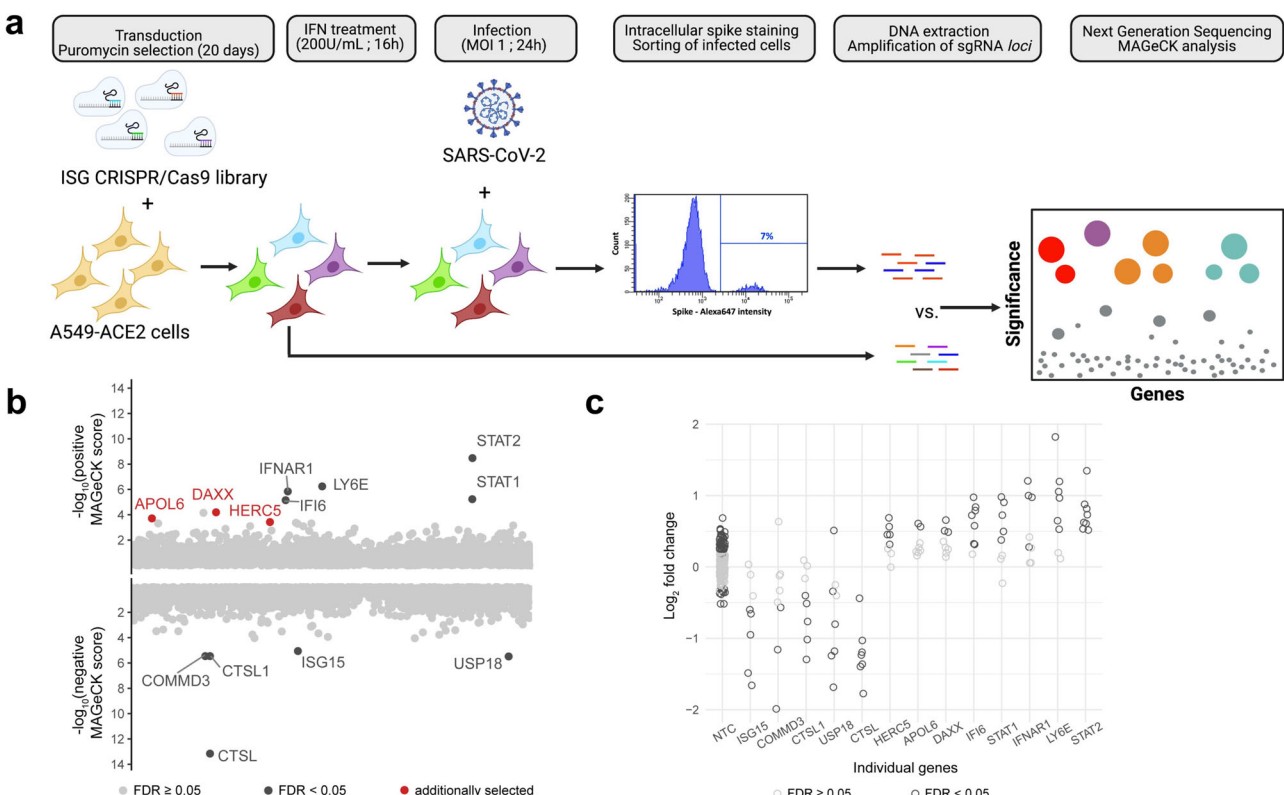

**Fig. 1 ISG-focused CRISPR/Cas9 screening approach to identify restriction factors for SARS-CoV-2.** **a** CRISPR/Cas9 screen outline. A549-ACE2 cells were transduced with lentivectors encoding the ISG CRISPR/Cas9 library and selected by puromycin treatment for 20 days. Library cells were then pre-treated with 200 U/mL of IFNα for 16 h, and infection with SARS-CoV-2 at an MOI of 1. At 24 h p.i., infected cells were fixed with formalin treatment, permeabilized by saponin treatment and stained with a monoclonal anti-spike antibody. After secondary staining, infected cells were sorted and harvested. Non-infected, non-IFNα treated cells were harvested as a control. DNA was extracted from both cellular fractions and sgRNA *loci* amplification was carried out by PCR. Following NGS, bio-informatic analysis using the MAGeCK package was conducted. This figure was created with BioRender.com. **b** Screen results. By taking into account the enrichment ratios of each of the 8 different sgRNAs for every gene, the MAGeCK analysis provides a positive score for KO enriched in infected cells (i.e. restriction factor, represented in the top fraction of the graph) and a negative score for KO depleted in infected cells (i.e. proviral factors, represented in the bottom portion of the graph). Genes with an FDR < 0.05 are represented in black. 3 genes with a FDR > 0.05, but with a *p* value < 0.005 were additionally selected and are represented in red. **c** Individual sgRNA enrichment. For the indicated genes, the enrichment ratio of the 8 sgRNAs present in the library was calculated as the MAGeCK normalized read counts in infected cells divided by those in the original pool of cells and is represented in log2 fold change. As a control, the enrichment ratio of the 200 non-targeting control (NTCs) is also represented.

electroporated with non-targeting sgRNAs/Cas9 RNPs (indicated here as WT). These different cell lines were then treated with IFNα and infected with SARS-CoV-2. Viral replication was assessed by measuring the levels of viral RNA in the supernatant of infected cells using RT-qPCR (Fig. 2a). In parallel, we titrated the levels of infectious viral particles released into the supernatant of infected cells (Fig. 2b). As expected, infection was significantly reduced in *CTSL* KO cells, confirming the proviral effect of this gene[36]. Among the selected antiviral candidate genes, only 2 had a significant impact on SARS-CoV-2 replication: *LY6E* (as expected), and to an even greater degree, *DAXX*. Both genes restricted replication in absence of IFNα, an effect which was detectable at the level of viral RNA (8-fold and 42-fold reduction of infection, respectively, Fig. 2a) and of infectious virus (15-fold and 62-fold reduction, Fig. 2b). Based on available single-cell RNAseq datasets,[43] *DAXX* is expected to be expressed in cell types physiologically relevant for SARS-CoV-2 infection such as lung epithelial cells and macrophages (Fig. S2).

In IFNα-treated cells, *DAXX* and *LY6E* KO led to a modest, but significant rescue of viral replication, which was particularly visible when measuring the levels of infectious virus by plaque assay titration (Fig. 2b), while the antiviral effect of IFNα treatment was completely abrogated in *IFNAR1* KO cells, as expected (Fig. 2c). However, IFNα still had robust antiviral effect on SARS-CoV-2 replication in both *DAXX* KO and *LY6E* KO cells (Fig. 2c). This

suggests that other ISGs likely contribute to the anti-SARS-CoV-2 IFN response. *DAXX* is sometimes referred to as an ISG, and was originally included in our ISG library, although its expression is only weakly induced by IFN in some human cell types[32,44]. Consistent with this, we found little to no increase in *DAXX* gene expression in IFNα-treated A549-ACE2 cells (Fig. S3). In addition, we tested the antiviral effect of DAXX on several SARS-CoV-2 variants that have been suggested to be partially resistant to the antiviral effect of IFN in A549-ACE2 cells[45]. Our results confirmed that Lineage B.1.1.7. (Alpha) and Lineage P1 (Gamma) SARS-CoV-2 variants were indeed less sensitive to IFN (Fig. 2d). DAXX, however, restricted all variants to a similar level than the original Lineage B strain of SARS-CoV-2 (Fig. 2d), suggesting that while some variants may have evolved towards IFN-resistance, they are still efficiently restricted by DAXX. To determine whether DAXX restriction is specific to SARS-CoV-2 or also inhibits other RNA viruses, including coronaviruses, A549-ACE2 WT and *DAXX* KO cells were infected with SARS-CoV, MERS-CoV, and 2 RNA viruses belonging to unrelated families: Yellow Fever Virus (YFV) and Measles Virus (MeV), which are positive and negative strand RNA viruses, respectively. DAXX restricted SARS-CoV, but had no effect on the replication of YFV, MeV or MERS-CoV (Fig. 2e, f). Thus, our data suggests that *DAXX* restriction may exhibit some level of specificity.

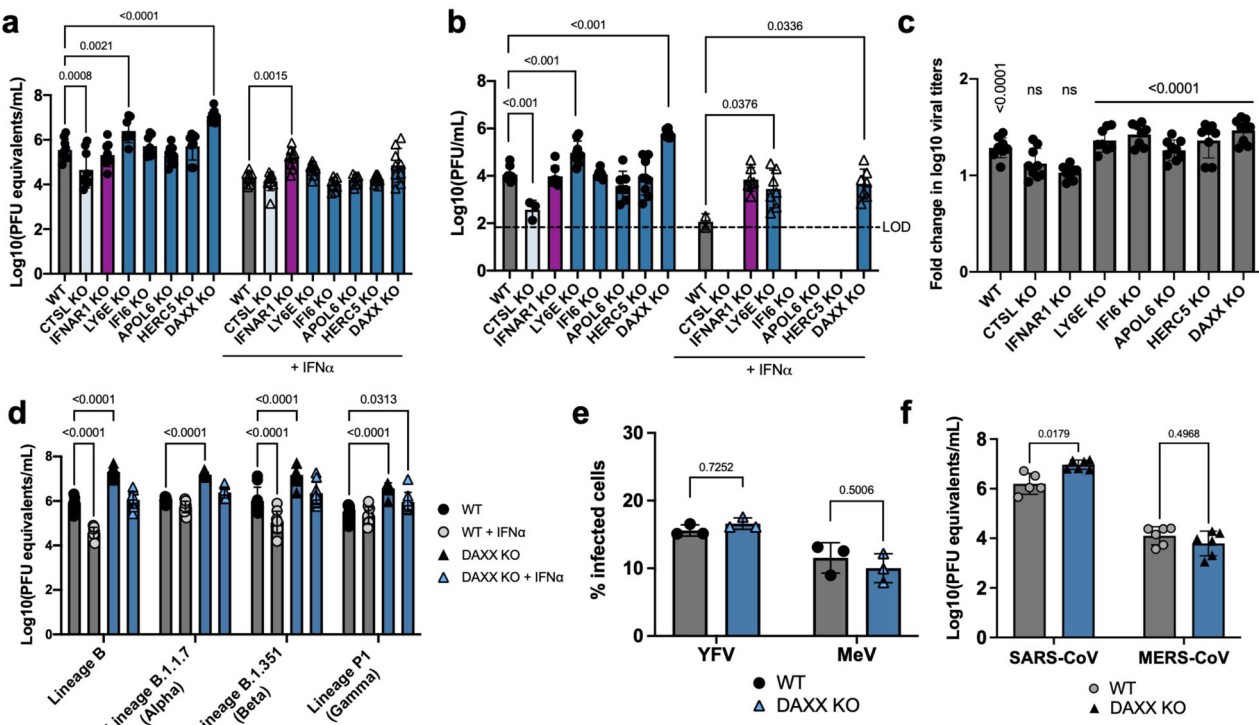

**Fig. 2 DAXX is a restriction factor for SARS-CoV-2. a–c** Antiviral activity of ISGs against SARS-CoV-2. A549-ACE2 knocked-out for the indicated genes were generated using a multi-guide approach, leading to pools of KO cells with a high frequency of indels. KO cells were pre-treated with 0 (circles) or 200 (triangles) U/mL of IFNα 24 h prior to infection with SARS-CoV-2 (at an MOI of 0.1). Supernatants were harvested at 72 h p.i. The mean ± SD of three independent experiments, each performed in three biological replicates, is shown. **a** For the titration of RNA levels, supernatants were heat-inactivated prior to quantification by qRT-PCR. Genome copies/mL were calculated by performing serial dilutions of a synthetic RNA with a known concentration. Statistics: 2-way ANOVA using Dunnett's test. Significant *p* values (below 0.05) are indicated on the graph. **b** For the titration of infectious virus levels by plaque assay, supernatants were serially diluted and used to infect VeroE6 cells. Plaques formed after 3 days of infection were quantified using crystal violet coloration. The limit of detection (LOD) is indicated as a dotted line. Statistics: Dunnett's test on a linear model, (two-sided). Significant *p* values (below 0.05) are indicated on the graph. **c** For each of the indicated KO, the data shown in (**a**) is represented as fold change in log10 titers (i.e. the log10 titers of the non-treated condition divided by the mean of the triplicate log10 titers IFNα-treated condition, n = 3). Statistics: 2-way ANOVA using Sidak's test. *P* values are indicated on the graph (ns: *p* value > 0.05). **d–f** Antiviral activity of DAXX against SARS-CoV-2 variants and other viruses. **d** A549-ACE2 WT or *DAXX* KO cells were infected at an MOI of 0.1 with the following SARS-CoV-2 strains: Lineage B (original strain); Lineage B.1.1.7. (Alpha variant); Lineage B.1.35.1 (Beta variant); Lineage P1 (Gamma variant). Supernatants were harvested at 72 h p.i. Supernatants were heat-inactivated prior to quantification by qRT-PCR. Genome copies/mL were calculated by performing serial dilutions of a synthetic RNA with a known concentration. The mean ± SD of three independent experiments, with infections carried out in three biological replicates, is shown. Statistics: 2-way ANOVA using Dunnett's test. Significant *p* values (below 0.05) are indicated on the graph. **e** A549-ACE2 WT or *DAXX* KO cells were infected with Yellow Fever Virus (YFV, Asibi strain, MOI of 0.3) or with Measles Virus (MeV, Schwarz strain expressing GFP, MOI of 0.2). At 24 h p.i., the percentages of cells positive for viral protein E (YFV) or GFP (MeV) was assessed by flow cytometry. The mean ± SD of 3 independent experiments is represented. Statistics: 2-way ANOVA using Sidak's test. *P* values are indicated on the graph. **f** WT or *DAXX* KO cells were infected at an MOI of 0.1 with SARS-CoV or MERS-CoV. Supernatants were harvested at 72 h p.i. Supernatants were heat inactivated prior to quantification by qRT-PCR. Serial dilutions of a stock of known infectious titer was used as a standard. The mean ± SD of 2 independent experiments, with infections carried out in three biological replicates, is represented. Statistics: 2-way ANOVA using Dunnett's test. *P* values are indicated on the graph. Source data are provided as a Source Data file.

**DAXX targets an early post-entry step**. Next, we investigated which steps of the SARS-CoV-2 replication cycle were targeted by DAXX. To assess whether DAXX affects SARS-CoV-2 Spike-mediated entry, we took advantage of a replication-competent Vesicular Stomatitis Virus expressing GFP (VSV*) and carrying the Spike protein instead of its G envelope (VSV*ΔG-S). This approach allows to study Spike-mediated viral entry without relying on lentiviral pseudotypes, which are likely to be targeted by DAXX[33]. We first ensured that DAXX was not affecting VSV replication in A549-ACE2 cells using the VSV* control virus (Fig. 3a). Flow cytometric analysis revealed that VSV*ΔG-S replicated at similar levels in WT and DAXX-KO cells (Fig. 3a), suggesting that *DAXX* does not inhibit the entry steps that are mediated by SARS-CoV-2 Spike.

To determine whether DAXX targets viral transcription, A549-ACE2 WT or *DAXX* KO cells were infected with SARS-CoV-2, and the intracellular levels of viral RNA were assessed at different time post-infection (Fig. 3c, d). At early time points (from 2 h to 6 h p.i.), the levels of viral RNA transcripts were similar in WT and *DAXX* KO cells, further suggesting that comparable amounts of SARS-CoV-2 RNA were entering cells in both cell lines. The levels of viral transcripts significantly increased starting at 8 h p.i., representing the initiation of viral transcription. The levels of viral RNA as detected by amplification of the 5′UTR (Fig. 3c) were 6.4-fold higher at 8 h; 4.1-fold higher at 10 h; and eightfold higher at 24 h post-infection in *DAXX* KO cells compared to WT cells. We observed a similar effect when using primers amplifying the RdRp region (Fig. 3d) with levels of viral transcripts 1.7-fold and 3.5-fold

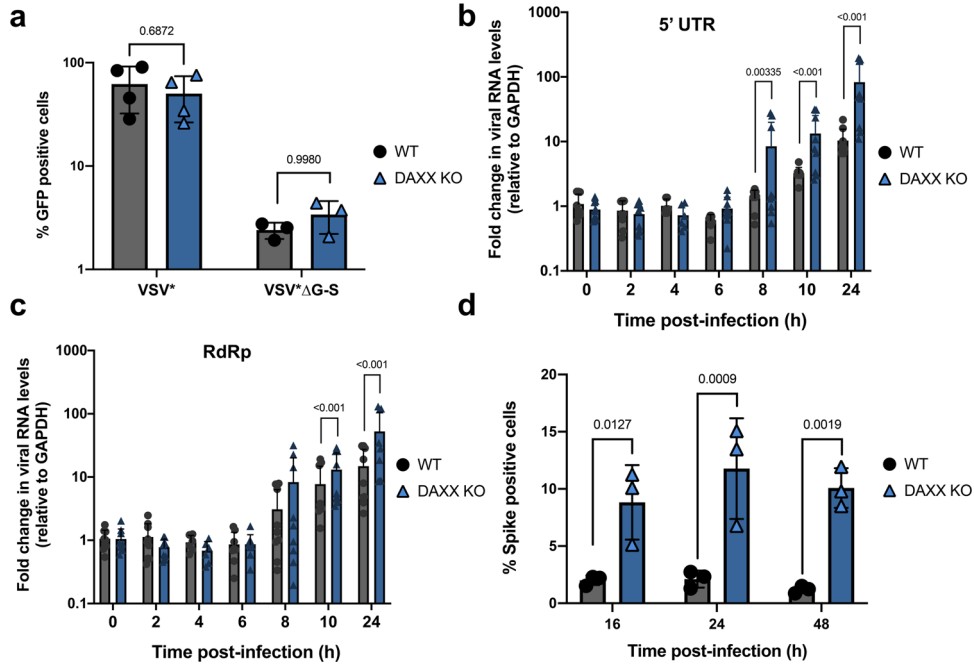

**Fig. 3 DAXX inhibits SARS-CoV-2 at an early post-entry step. a** DAXX has no effect on Spike-mediated entry. A549-ACE2 WT or *DAXX* KO were infected with the GFP reporter viruses VSV* or VSV*ΔG-S. Cell monolayers were harvested at 16 h p.i. and analyzed by flow cytometry. The mean of 3 independent experiments ± SD is shown. Statistics: 2-way ANOVA using Sidak's test. *P* values are indicated on the graph. **b**, **c** DAXX inhibits viral RNA synthesis. A549-ACE2 WT or *DAXX* KO were infected at an MOI of 1. Cell monolayers were harvested at the indicated time points, and total RNA was extracted. The levels of viral RNA (**c**: 5′UTR; **d**: RdRp) were determined by qRT-PCR and normalized against GAPDH levels. The mean ± SD of 3 independent experiments, with infections carried out in three biological replicates, is represented. Statistics: Dunnett's test on a linear model (two-sided). Significant *p* values (below 0.05) are indicated on the graph. **d** DAXX inhibits viral protein synthesis. A549-ACE2 WT or *DAXX* KO were infected at an MOI of 2. Cell monolayers were harvested at the indicated time points and Spike levels were quantified by flow cytometry. The mean of 3 independent experiments ± SD is represented. Statistics: 2-way ANOVA using Sidak's test. *P* values are indicated on the graph. Source data are provided as a Source Data file.

higher in *DAXX* KO cells compared to WT cells at 10 h and 24 h post-infection, respectively.

Finally, we measured the levels of Spike protein synthesis at different times post-infection by flow cytometry in A549-ACE2 WT or *DAXX* KO cells. In agreement with the observed effect of *DAXX* KO on viral RNA synthesis (Fig. 3c, d), we observed a 4- to 8-fold increase in the intracellular levels of Spike (Fig. 3e). Together, these results suggest that while DAXX has no effect on viral entry, it restricts a post-entry step of the viral life cycle such as viral transcription or uncoating.

**DAXX restriction is mediated by its D/E domain but is SUMOylation-independent.** *DAXX* encodes a small scaffold protein that acts by recruiting other SUMOylated proteins in nuclear bodies through its C-terminal SUMO-Interacting Motif (SIM) domain[46]. The recruitment of these factors is required for the effect of DAXX on various cellular processes such as transcription and apoptosis, and on its antiviral activities[32,47–49]. DAXX can also be SUMOylated itself[50], which may be important for some of its functions. To investigate the role of SUMOylation in DAXX-mediated SARS-CoV-2 restriction, overexpression assays using WT and two previously described mutated versions of DAXX[51] were performed (Fig. 4a). Fifteen lysine residues have been mutated to arginine in the first mutant (DAXX 15KR), which is unable to be SUMOylated. The second mutant is a truncated version of DAXX that lacks its C-terminal SIM domain (DAXXΔSIM)[48] and is thus unable to interact with its SUMOylated partners. A549-ACE2 were refractory to SARS-CoV-2 infection upon transfection with any plasmid, precluding us from using this cell line. The experiments were performed in

293T-ACE2 cells, which are permissive to SARS-CoV-2[18] and easy to transfect. In order to quantify the antiviral effect of overexpressed DAXX WT and mutants, we assessed the number of transfected cells that were positive for the Spike protein by flow cytometry. Western blot (Fig. S4a) and flow cytometry (Fig. S4b) analyses showed that DAXX WT and mutants were expressed at similar levels, with around 40 to 50% cells expressing the HA-tagged constructs. DAXX WT, 15KR and ΔSIM efficiently restricted SARS-CoV-2 replication. Indeed, at 24 h p.i., the proportion of infected cells (among HA-positive cells) was reduced by 2 to 3-fold as compared to control transfected cells (Fig. 4b). This effect was less pronounced but still significant at 48 h p.i. (Fig. 4c). Moreover, DAXX overexpression led to a significant reduction of the levels of intracellular viral RNA (Fig. S5), in line with our earlier results showing that DAXX targets viral transcription (Fig. 3c, d). Together, these results show that DAXX overexpression restricts SARS-CoV-2 replication in a SUMOylation-independent mechanism.

DAXX was recently described as a protein chaperone able to solubilize protein aggregates and unfold misfolded proteins[52]. We investigated whether this activity was required for SARS-CoV-2 restriction using a DAXX mutant lacking the D/E domain, critical for this chaperone activity[52]. The DAXXΔD/E mutant, while expressed at similar levels than WT DAXX in transfected cells (Fig. S4c), was unable to restrict SARS-CoV-2, as assessed by Western Blot analysis on Spike levels (Fig. 4d) and RT-qPCR analysis detecting intracellular SARS-CoV-2 RNA (Fig. 4e). Taken together, these results suggest that DAXX dampens SARS-CoV-2 replication through a SUMOylation-independent mechanism that likely involves its chaperone activity.

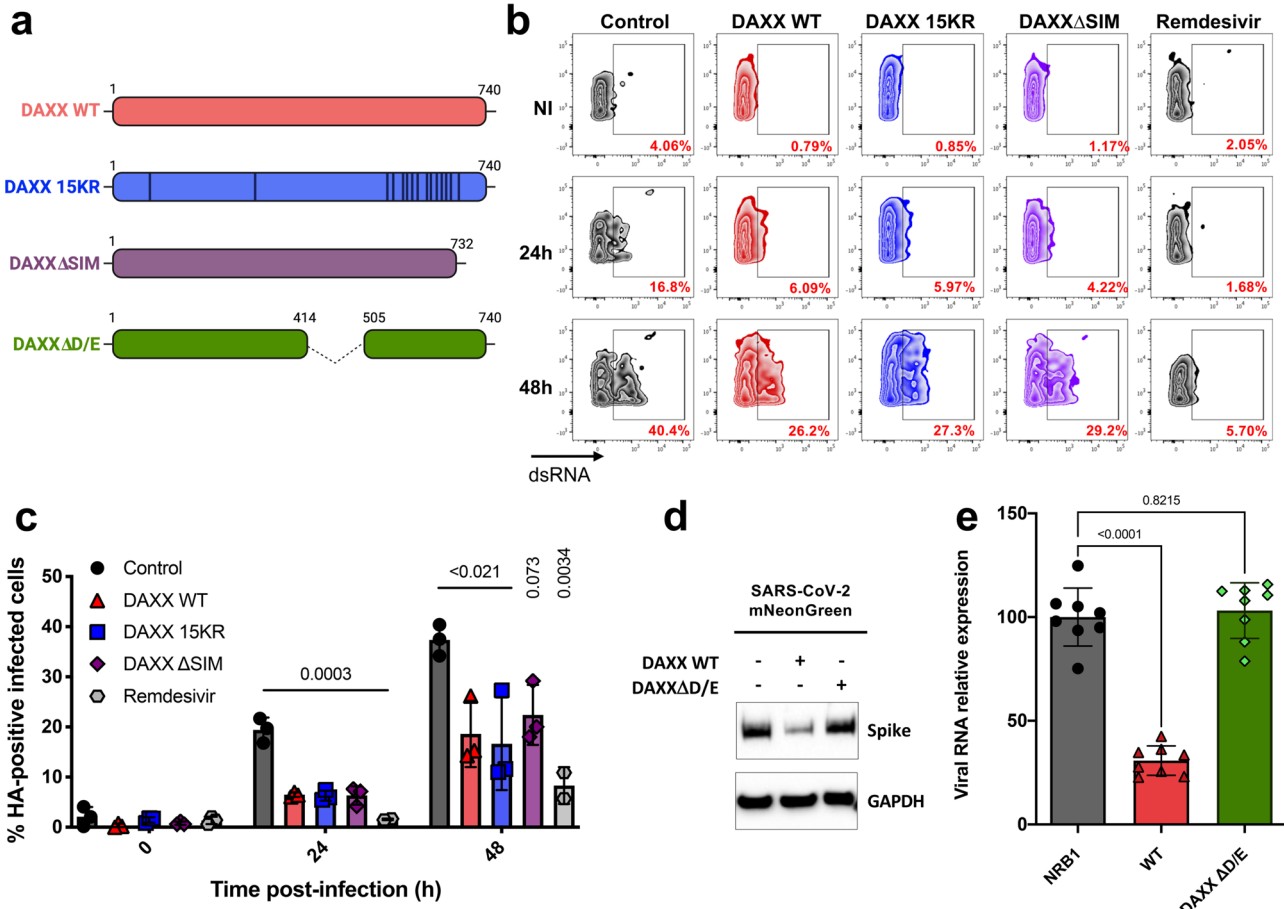

**Fig. 4 DAXX restriction of SARS-CoV-2 is dependent on its chaperone activity but SUMOylation-independent. a** Schematic of the DAXX mutants used. The fifteen lysine residues of DAXX 15KR have been mutated to arginine. DAXXΔSIM lacks the 732–740 C-terminal residues. Both mutants were described in.[48] DAXXΔD/E is lacking its 414-505 domain and has been described in[52] **b**, **c** SUMOylation-deficient DAXX mutants are still able to restrict SARS-CoV-2. 293T-ACE2 cells were transfected with HA-DAXX WT; HA-DAXX 15KR; HA-DAXXΔSIM; or with HA-NBR1 as negative control plasmid. 24 h after transfection, cells were infected with SARS-CoV-2 at an MOI of 0.1. When indicated, cells were treated with remdesivir at the time of infection. After 24 h or 48 h, infected cells were double-stained for dsRNA (to read out infection) and HA (to read out transfection efficiency) and acquired by flow cytometry. The percentage of infected cells among HA-positive (transfected) cells for one representative experiment is shown in (**b**), for the mean ± SD of 3 independent experiments in (**c**). Statistics: one-way ANOVA using Dunnett's test, Holm corrected. *P* values are indicated on the graph. **d**, **e**: The chaperone activity of DAXX is required for SARS-CoV-2 restriction. 293T-ACE2 cells were transfected with DAXX WT or with the DAXXΔD/E mutant. 24 h after transfection, cells were infected with SARS-CoV-2 mNeonGreen at an MOI of 1. After 24 h, the levels of Spike and GAPDH levels were analyzed by Western Blot (**d**). A Western Blot representative of 3 independent experiments is shown. In parallel, SARS-CoV-2 replication levels were measured by RT-qPCR targeting the 5′ UTR and normalized against RPL13A transcript levels (**e**). The mean ± SD of 4 independent experiments, with infections carried out in two biological replicates, is represented. Statistics: 1-way ANOVA using Dunnett's test. *P* values are indicated on the graph. Source data are provided as a Source Data file.

**SARS-CoV-2 infection triggers DAXX re-localization.** DAXX mostly localizes in PML nuclear bodies[30], whereas SARS-CoV-2 replication occurs in the cytoplasm. We reasoned that DAXX may re-localize during the course of infection in order to exert its antiviral effect. We first examined the effect of DAXX over-expression on the replication of SARS-CoV-2-mNeonGreen[53] by microscopy. DAXX overexpression starkly reduced the number of infected cells (Fig. 5a, b), confirming our flow cytometry data (Fig. 4). Using double staining for HA-tagged DAXX and SARS-CoV-2, we found that most of the DAXX-transfected cells were negative for infection, and conversely, that most of the infected cells did not express transfected DAXX (Fig. 5c), confirming that DAXX imposes a major block to SARS-CoV-2 infection. Next, we infected 293T-ACE2 cells with SARS-CoV-2 and used high-resolution confocal microscopy to study the localization of endogenous DAXX (Fig. 5d). As expected[30], DAXX localized in discrete nuclear *foci* in non-infected cells. Strikingly, SARS-CoV-2 infection induced the re-localization of DAXX in the cytoplasm,

as early as 6 h post-infection, although some nuclear *foci* were still detected. At 24 h post-infection however, DAXX was completely absent from nuclear bodies, and was found almost exclusively in the cytoplasm, in close association with dsRNAs, likely representing SARS-CoV-2 replication sites. These results suggest that early events following SARS-CoV-2 infection trigger the cytoplasmic translocation of DAXX.

**SARS-CoV-2 PLpro induces proteasomal degradation of DAXX.** Next, we asked whether this relocalization of DAXX following infection destabilizes the protein. Western blot analysis revealed that SARS-CoV-2 infection induces a marked decrease of total DAXX expression levels in infected cells (Fig. 6a). In contrast, SARS-CoV-2 infection had no effect on *DAXX* mRNA levels (Fig. S6). Importantly, the decrease in DAXX protein levels is likely not attributed to a global host protein expression shut down, as the levels of Lamin B, HSP90, Actin, GAPDH, Tubulin, TRIM22 and RIG-I were unchanged upon infection (Fig. 6a).

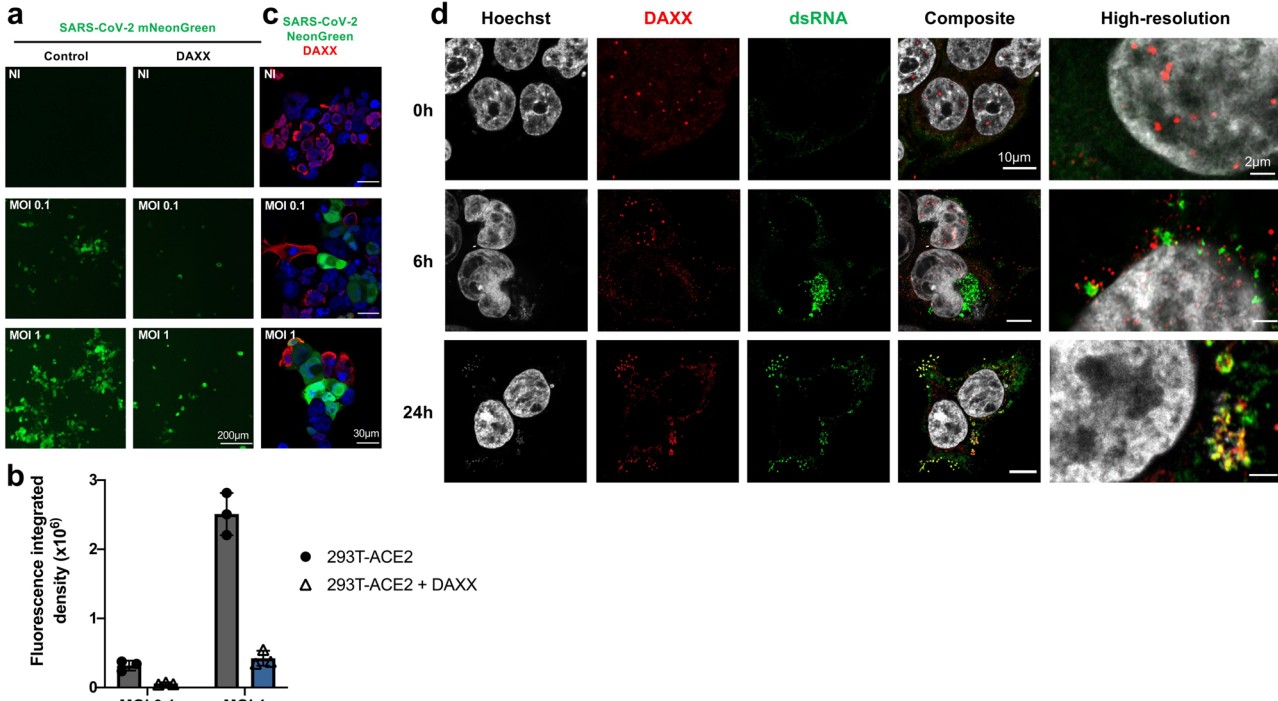

**Fig. 5 SARS-CoV-2 infection induces DAXX cytoplasmic re-localization to sites of viral replication.** DAXX overexpression restricts SARS-CoV-2. 293T-ACE2 cells were transfected with DAXX WT. 24 h after transfection, cells were infected with the mNeonGreen fluorescent reporter SARS-CoV-2 at the indicated MOI. Cells were either visualized with an EVOS fluorescence microscope (**a**, **b**) or stained with an HA-antibody detecting DAXX and imaged by confocal microscopy (**c**). Scale bars correspond to 200 μm (**a**) and 30 μm (**c**). Images shown in (**a**) were quantified using ImageJ software (**b**). Data shows the mean ± SD of Fluorescence integrated densities. The analysis was performed on around 200 cells from 3 different fields. Images are representative of 3–6 different fields from 2 independent experiments. **d** Relocalization of endogenous DAXX during SARS-CoV-2 infection. 293T-ACE2 cells were infected with SARS-CoV-2 at an MOI of 1. 24 h post-infection, cells were labelled with Hoescht and with antibodies against dsRNA (detecting viral RNA, in green) and HA (detecting DAXX, in red). When indicated, the high-resolution Airyscan mode was used. Scale bars correspond to 10 μm for confocal images, and 2 μm for the high-resolution images. Images are representative of 3–6 different fields from 2 independent experiments. Source data are provided as a Source Data file.

These results suggest that DAXX may be specifically targeted by SARS-CoV-2 for degradation. SARS-CoV-2 papain-like protease (PLpro) is a possible candidate for this activity, as it cleaves other cellular proteins such as ISG15[54,55], and ULK1[56]. Moreover, PLpro of foot-and-mouth disease virus (FDMV) degrades DAXX[57]. To investigate this possibility, we treated cells with GRL0617, an inhibitor of SARS-CoV-2 PLpro[55]; MG132, a well-described proteasome inhibitor; or Masitinib, an inhibitor of SARS-CoV-2 3CL protease[58]. These inhibitors had minimal effects on cell viability at the selected concentrations (Fig. S7). Strikingly, GRL0617 treatment partially restored DAXX expression (Fig. 6b), especially at the highest concentration. Similarly, MG132 also prevented DAXX degradation in SARS-CoV-2 infected cells. In contrast, Masitinib treatment had no effect on DAXX levels. These results suggest that PLpro, but not 3CL, targets DAXX for proteasomal degradation. Consistently, GRL0617 treatment also restored DAXX subcellular localization to nuclear bodies (Fig. 6c). As expected, GRL0617 treatment also inhibited the production of SARS-CoV-2 proteins, such as Spike (Fig. 6b), and may thus have an indirect effect on DAXX levels by inhibiting SARS-CoV-2 replication itself. However, the fact that Masitinib also inhibits Spike production but does not restore DAXX expression suggested that DAXX degradation is not an unspecific consequence of a reduced viral replication but rather a specific activity of PLpro. To investigate further the direct contribution of PLpro to DAXX degradation, we assessed the impact of overexpressing a panel of individual SARS-CoV-2 proteins in 293T-ACE2 cells on DAXX levels. We included in the analysis

mCherry-tagged SARS-CoV-2 Non-structural proteins (Nsp[59], which were not expressed from a lentiviral vector that may be targeted by DAXX antiviral activity[33]. This included Nsp3 (which encodes PLro), Nsp4, Nsp6, Nsp7, Nsp10, Nsp13 and Nsp14. All proteins were expressed at similar levels (Fig. S8a). Only the overexpression of Nsp3 led to DAXX reduced expression (Fig. 6d, Fig. S8b). This effect was dose-dependent (Fig. 6e, Fig. S8c, d), and was abrogated when cells were treated with GRL0617 (Fig. 6f, Fig. S8e). Taken together, these results indicate that PLpro directly induces the proteasomal degradation of DAXX.

## Discussion

The whole-genome CRISPR/Cas9 screens conducted to date on SARS-CoV-2 infected cells mostly identified host factors necessary for viral replication[24–29] and did not focus on antiviral genes, as did our screen. Three overexpression screens, however, identified ISGs with antiviral activity against SARS-CoV-2[16,21,22]. In the first one, Pfaender et al. screened 386 ISGs for their antiviral activity against the endemic human coronavirus 229E, and identified *LY6E* as a restriction factor inhibiting both 229E and SARS-CoV-2. Our screen also identified *LY6E* as a top hit (Fig. 1), further validating the findings of both studies. Four additional genes had significant *p*-values in both Pfaender et al. and our work: *IFI6*, *HERC5*, *OAS2* and *SPSB1* (Supplementary Data 1 and 2). We showed that knocking-out *LY6E* and *DAXX* only partially rescued SARS-CoV-2 replication in IFN-treated cells (Fig. 2), suggesting that other IFN effectors active against SARS-CoV-2 remain to be identified. For instance, other proteins, such as

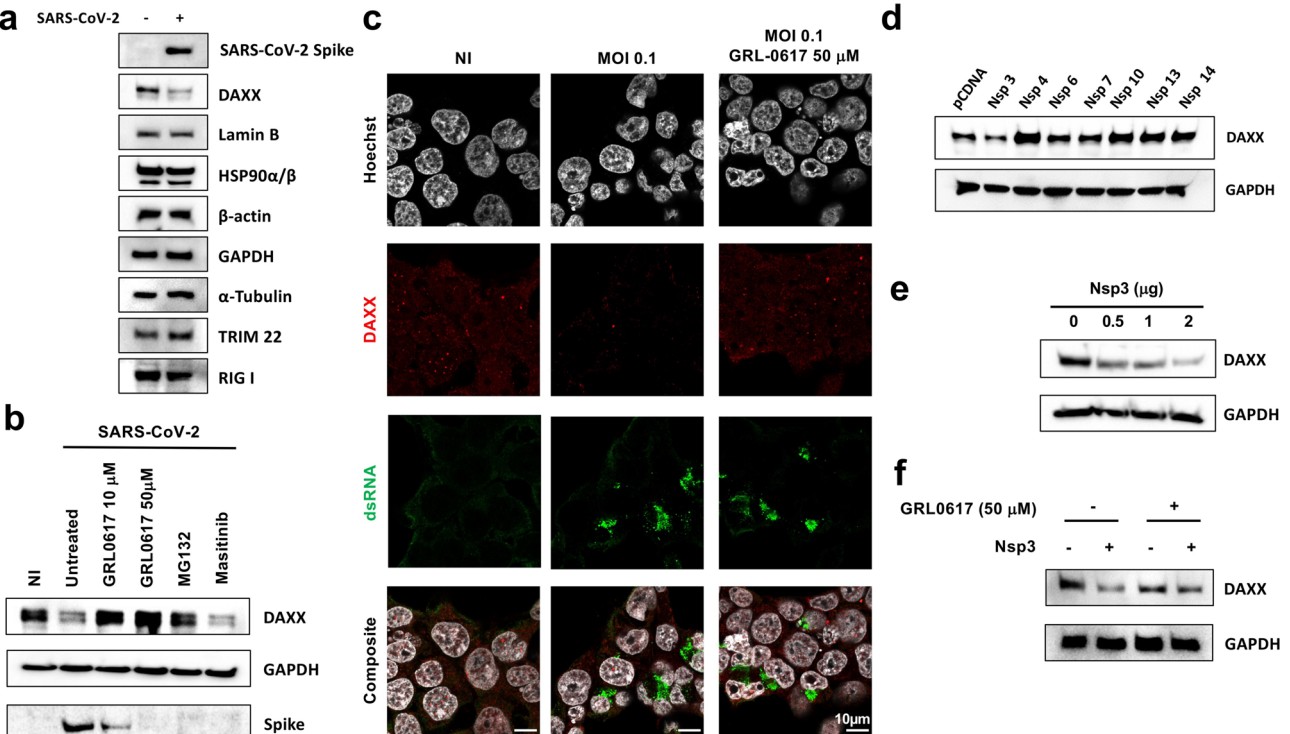

**Fig. 6 SARS-CoV-2 PLpro induces the proteasomal degradation of DAXX.** **a** DAXX degradation after infection. 293T-ACE2 cells were infected with SARS-CoV-2 at MOI 0.1. After 24 h, cells were harvested and levels of DAXX, Lamin B, HSP90, Actin, GAPDH, Tubulin, TRIM22, RIG-I and of the viral protein spike were analyzed by Western Blot. A Western Blot representative of 3 independent experiments is shown. **b** GRL0617 and MG132 treatments restore DAXX expression. 293T-ACE2 cells were infected with SARS-CoV-2 at an MOI of 0.1. When indicated, cells were pretreated 2 h before infection with GRL0617 (at the indicated concentrations), or with MG132 (10 μM), a proteasome inhibitor, or Masitinib (10 μM) a 3CL inhibitor. After 24 h, cells were harvested and levels of DAXX, GAPDH and of the viral protein spike were analyzed by Western Blot. A Western Blot representative of 3 independent experiments is shown. **c** GRL0617 treatment restores DAXX localization. 293T-ACE2 cells were infected with SARS-CoV-2 at an MOI of 0.1. 24 h post-infection, cells were labelled with Hoescht and with antibodies against dsRNA (detecting viral RNA, in green) and HA (detecting DAXX, in red). When indicated, cells were treated with 50 μM of GRL0617 at the time of infection. Scale bars correspond to 10 μm. Images are representative of 3–6 different fields from 2 independent experiments. **d–f**: Nsp3 induces DAXX degradation. **d** 293T-ACE2 cells were transfected with 1 μg of the indicated viral proteins. After 24 h, the levels of DAXX and GAPDH were analyzed by Western Blot. **e** 293T-ACE2 cells were transfected with the indicated amounts of Nsp3. After 24 h, the levels of DAXX and GAPDH were analyzed by Western Blot. **f** 293T-ACE2 cells were transfected with 1 μg of Nsp3 or of pcDNA. 6 h post transfection, cells were also, when indicated, treated with 50 μM of GRL0617. Of, 24 h after transfection, the levels of DAXX and GAPDH were analyzed by Western Blot. Western Blots representative from 2 independent experiments are shown. The quantification of band intensity for Fig. 6d–f is shown in Fig. S8b,d, e. Source data are provided as a Source Data file.

IFITMs, inhibit SARS-CoV-2 viral entry[17–19]. In the second screen, Martin Sancho et al. tested 399 ISGs against SARS-CoV-2. Among the 65 antiviral ISGs identified, they focused on *BST-2*, which encodes a protein targeting viral budding. *BST-2* was not a significant hit in our screen (Supplementary Data 1 and 2). This discrepancy is likely due to the fact that our screen relies on the sorting of S-positive cells, and is therefore unable to detect factors restricting the late stages of the viral replication cycle. The most recent overexpression screen assessed the contribution of 539 human and 444 macaque ISGs in SARS-CoV-2 restriction, and further characterized the role of *OAS1* in sensing SARS-CoV-2 and restricting its replication through *RNASEL*. While we did not identify *OAS1* or *RNASEL* in our screen (Supplementary Data 1 and 2), we did identify hits in common with this screen, including *IFI6* and *OAS2* (that were also identified by Pfaender et al.). Of note, *DAXX* was absent from the ISG libraries used by these overexpression screens, which explains why it was not previously identified as an antiviral gene for SARS-CoV-2. Our sgRNA library, by including 1905 genes, targeted a wider set of ISGs and "ISG-like" genes, including genes like *DAXX* that are not (or only weakly) induced by IFN in some cell types[32,44]. One potential caveat to our screen is that it compared IFN-treated infected cells

to non-infected untreated cells (rather than IFN-treated non-infected cells). Although we validated this approach in previous studies[34,60], it may cause enrichment of genes induced by IFN, but not antiviral against SARS-CoV-2 per se. Interestingly, IFN has a stronger effect on *DAXX* gene expression levels in cells from other mammals such as bats.[61] Future studies may investigate whether *DAXX* orthologs of different species are also able to restrict SARS-CoV-2 and whether *DAXX* participates in IFN-mediated viral restriction in these species.

We identify DAXX as a potent antiviral factor restricting the replication of SARS-CoV-2, acting independently of IFN (Fig. 3). DAXX fulfills all of the criteria defining a bona fide SARS-CoV-2 restriction factor: knocking-out endogenous DAXX leads to enhanced viral replication (Fig. 2), while over-expression of DAXX restricts infection (Figs. 4–5). While DAXX had no effect on Spike-mediated viral entry (Fig. 3), it led to a stark reduction in the levels of viral transcripts at 8 h post-infection, in the levels of Spike protein at 16 h post-infection (Fig. 3). This suggests that DAXX blocks a post-entry step of the viral life cycle such as viral transcription. DAXX co-localizes with viral replication sites (Fig. 5) and SARS-CoV-2 antagonizes DAXX to some extent, as evidenced by the proteasomal degradation of DAXX induced by

PLpro (Fig. 6). Although *DAXX* expression is not upregulated by IFNα in A549 cells (Fig. S3), basal levels of expression are sufficient for its antiviral activity, as has been shown for other potent restriction factors. Publicly available single-cell RNAseq analyses (Fig. S2) indicated that *DAXX* is expressed in cell types targeted by the virus in vivo, such as lung epithelial cells and macrophages. Interestingly, DAXX exhibited some degree of specificity in its antiviral activity, as unrelated viruses such as YFV and MeV, as well as the closely related MERS-CoV were not sensitive to its action, in contrast to SARS-CoV and SARS-CoV-2 (Fig. 2). Future work will determine which viral determinants are responsible for the specific antiviral activity of DAXX.

DAXX is mostly known for its antiviral activity against DNA viruses replicating in the nucleus, such as adenovirus 5 (AdV5)[62] and human papillomavirus (HPV)[63]. Most of these viruses antagonize *PML* and/or DAXX, which interacts with *PML* in nuclear bodies[30]. We show here that DAXX is also able to restrict SARS-CoV-2, a positive sense RNA virus that replicates in the cytoplasm. Recent studies have shown that DAXX inhibits the reverse transcription of HIV-1 in the cytoplasm[32,33]. Within hours of infection, DAXX subcellular localization was altered, with DAXX accumulating in the cytoplasm and colocalizing with incoming HIV-1 capsids[33]. Here, we observed a similar phenomenon, with a rapid re-localization of DAXX from the nucleus to cytoplasmic viral replication sites (Fig. 5), where it likely exerts its antiviral effect. Early events in the replication cycle of both HIV-1 and SARS-CoV-2, such as fusion between viral and cellular membranes, or virus-induced stress, may thus trigger DAXX re-localization to the cytoplasm. DAXX seems to inhibit SARS-CoV-2 by a distinct mechanism than HIV-1: whereas the recruitment of interaction partners through the SIM-domain is required for the effect of DAXX on HIV-1 reverse transcription[32], it was not the case in the context of SARS-CoV-2 restriction. This result was unexpected, since DAXX has no enzymatic activity and rather acts as a scaffold protein recruiting SUMOylated partners through its SIM domain[51]. Other DAXX functions, such as interaction with the chromatin remodeler ATRX[30] or its role as a chaperone protein[52] are, however, SIM-independent. This last activity was recently shown[52] to be dependent on the D/E domain (amino acids 414 to 505). We found that DAXX anti-SARS-CoV-2 activity also depends on this domain (Fig. 4), suggesting that it restricts SARS-CoV-2 replication through an original mechanism involving protein refolding. Future work will determine whether DAXX binds and refolds viral proteins to hamper viral replication, or acts through binding and folding another host factor.

Our results suggest that SARS-CoV-2 evolved a mechanism to antagonize DAXX restriction, with PLpro inducing its degradation via the proteasome (Fig. 6). This antagonism, however, is not complete, since knocking-out DAXX expression enhanced SARS-CoV-2 replication (Fig. 2). Another possibility is that DAXX, by acting early in the viral life cycle (i.e. as soon as 8 h p.i., Fig. 3) exert its antiviral effect before the expression of PLpro. Proteins expressed by other viruses are also able to degrade DAXX: for instance, the AdV5 viral factor E1B-55K targets DAXX for proteasomal degradation[62], and FDMV PLpro cleaves DAXX[57]. We showed in Fig. 2 that several SARS-CoV-2 variants and SARS-CoV were sensitive to DAXX, but MERS-CoV was not. Thus, it will be interesting to test whether PLpro from these different coronaviruses differ in their ability to degrade DAXX, and whether this has an impact on their sensitivity to DAXX restriction. Future research may also establish whether PLpro induces the degradation of DAXX through direct cleavage, or whether it acts in a more indirect way, such as cleaving or recruiting cellular co-factors. Such investigations may be relevant for the development of PLpro inhibitors[64]: indeed, in addition to directly blocking

SARS-CoV-2 replication, PLpro inhibitors may also sensitize the virus to existing antiviral mechanisms such as DAXX restriction.

## Methods

**Cells, viruses & plasmids**. HEK 293T (ATCC #CRL-11268) were cultured in MEM (Gibco #11095080) complemented with 10% FBS (Gibco #A3160801) and 2 mM L-Glutamine (Gibco # 25030081). VeroE6 (ATCC #CRL-1586), A549 (ATCC #CCL-185) and HEK 293T, both overexpressing the ACE2 receptor (A549-ACE2 and HEK 293T-ACE2, respectively), were grown in DMEM (Gibco #31966021) supplemented with 10% FBS (Gibco #A3160801), and penicillin/streptomycin (100 U/mL and 100 μg/mL, Gibco # 15140122). Blasticidin (10 μg/mL, Sigma-Aldrich #SBR00022-10ML) was added for selection of A549-ACE2 and HEK 293T-ACE2. All cells were maintained at 37 °C in a 5% $CO_2$ atmosphere. Universal Type I Interferon Alpha (PBL Assay Science #11200-2) was diluted in sterile-filtered PBS 1% BSA according to the activity reported by the manufacturer. The strains BetaCoV/France/IDF0372/2020 (Lineage B); hCoV-19/France/IDF-IPP11324/2020 (Lineage B.1.1.7); and hCoV-19/France/PDL-IPP01065/2021 (Lineage B.1.351) were supplied by the National Reference Centre for Respiratory Viruses hosted by Institut Pasteur and headed by Pr. Sylvie van der Werf. The human samples from which the lineage B, B.1.1.7 and B.1.351 strains were isolated were provided by Dr. X. Lescure and Pr. Y. Yazdanpanah from the Bichat Hospital, Paris, France; Dr. Besson J., Bioliance Laboratory, saint-Herblain France; Dr. Vincent Foissaud, HIA Percy, Clamart, France, respectively. These strains were supplied through the European Virus Archive goes Global (Evag) platform, a project that has received funding from the European Union's Horizon 2020 research and innovation programme under grant agreement #653316. The hCoV-19/Japan/TY7-501/2021 strain (Lineage P1) was kindly provided by Jessica Vanhomwegen (Cellule d'Intervention Biologique d'Urgence; Institut Pasteur). The mNeonGreen reporter SARS-CoV-2 was provided by Pei-Yong Shi[53]. SARS-CoV FFM-1 strain[65] was kindly provided by H.W. Doerr (Institute of Medical Virology, Frankfurt University Medical School, Germany). The Middle East respiratory syndrome (MERS) Coronavirus, strain IP/COV/MERS/Hu/France/FRA2 (Genbank reference KJ361503) isolated from one of the French cases[66] was kindly provided by Jean-Claude Manuguerra (Cellule d'Intervention Biologique d'Urgence; Institut Pasteur). SARS-CoV-2 viral stocks were generated by infecting VeroE6 cells (MOI 0.01, harvesting at 3 dpi) using DMEM supplemented with 2% FBS and 1 μg/mL TPCK-trypsin (Sigma-Aldrich #1426-100MG). SARS-CoV and MERS-CoV viral stocks were generated by infecting VeroE6 cells (MOI 0.0001) using DMEM supplemented with 5% FCS and harvesting at 3 dpi or 6 dpi, respectively. The Yellow Fever Virus (YFV) Asibi strain was provided by the Biological Resource Center of the Institut Pasteur. The Measles Schwarz strain expressing GFP (MeV-GFP) was described previously.[67] Both viral stocks were produced on Vero NK cells. The Human Interferon-Stimulated Gene CRISPR Knockout Library was a gift from Michael Emerman and is available on Addgene (Pooled Library #125753). The plentiCRISPRv.2 backbone was ordered through Addgene (Plasmid #52961). pMD2.G and psPAX2 were gifts from Didier Trono (Addgene #12259; #12260). pcDNA3.1 was purchased from Invitrogen. Plasmids constructs expressing WT and mutant HA-tagged DAXX constructs were kindly provided by Hsiu-Ming Shih[51]. The plasmids encoding mCherry-tagged viral proteins were a gift from Bruno Antonny and ordered through Addgene: Nsp3 -mCherry (#165131); Nsp4-mCherry (#165132); Nsp6-mCherry (#165133); Nsp7-mCherry (#165134); Nsp10-mCherry (#165135); Nsp13-mCherry (#165136); Nsp14-mCherry (#165137). Vesicular stomatitis virus (VSV) encoding green fluorescent protein (GFP) has been previously described as VSV*[68]. The chimeric virus VSV*ΔG-SARS-CoV-2-S$_{\Delta21}$ (VSV*ΔG-S), which lacks the homotypic glycoprotein G but rather encodes the spike protein of SARS-CoV-2 (Wuhan-Hu-1 strain) along with GFP has recently been described[69]. The DAXXΔD/E mutant is a kind gift from Pr. Xiaolu Yang.

**Antibodies**. For Western Blot, we used mouse anti-DAXX (diluted 1:1000, Abnova #7A11), rat anti-HA clone 3F10 (diluted 1:3000, Sigma #2158167001), mouse anti-GAPDH clone 6C5 (diluted 1:3000, Millipore #FCMAB252F), Goat anti-Lamin B clone M-20 (diluted 1:500, Santa Cruz sc-6217), mouse monoclonal HSP90α/β clone F-8 (diluted 1 :500, Santa Cruz sc-13119), mouse monoclonal β-actin clone AC-15 (1:3000 Sigma #A1978), mouse monoclonal α-Tubulin clone DMA1 (diluted 1:1000, Sigma #T9026), rabbit anti-TRIM22 (diluted 1 :1000, Proteintech #13744-1-AP) and mouse Monoclonal RIG-I clone Alme-1 (diluted 1: 1000, adipoGen #AG-20B-0009). To detect SARS-CoV-2 Spike protein, we used mouse anti-spike clone 1A9 (diluted 1:1000, GeneTex GTX632604). Secondary antibodies were goat anti-mouse and anti-rabbit HRP-conjugates (diluted 1:5000, ThermoFisher #31430 and #31460) and horse anti-goat HRP (diluted 1: 1000, Vector * PI-9500). For immunofluorescence, we used rabbit anti-DAXX (diluted 1:50, Proteintech #20489-1-AP) and mouse anti-dsRNA J2 (diluted 1:50, Scicons #10010200). Secondary antibodies were goat anti-rabbit AF555 and anti-mouse AF488 (diluted 1:1000, ThermoFisher #A-21428 and #A-28175). For flow sorting of infected cells, we used the anti-S2 H2 162 antibody (diluted 1:150), a kind gift from Dr. Hugo Mouquet, (Institut Pasteur, Paris, France). Secondary antibody was donkey anti-mouse AF647 (diluted 1:1000, Invitrogen #A31571). For FACS analysis, we used rat anti-HA clone 3F10 (diluted 1:100, Sigma #2158167001) and mouse anti-dsRNA J2 (diluted 1:500, Scicons #10010200).

Secondary antibodies were goat anti-rat AF647 and anti-mouse AF488 (diluted 1:1000, ThermoFisher #A-21247 #A-28175). The pan-flavivirus anti-Env 4G2 antibody was a kind gift from Phillipe Desprès.

**Generation of CRISPR/Cas9 library cells**. HEK 293T cells were transfected with the sgRNA plasmid library together with plasmids coding for Gag/Pol (R8.2) and for the VSVg envelope (pVSVg) using a ratio of 5:5:1 and calcium phosphate transfection. Supernatants were harvested at 36 h and 48 h, concentrated 80-fold by ultracentrifugation (22,000 g, 4 °C for 1 h) and pooled. To generate the ISG KO library cells, $36 \times 10^6$ A549-ACE2 cells were seeded in 6-well plates ($10^6$ cells per well) 24 h before transduction. For each well, 100 μL of concentrated lentivector was diluted in 500 μL of serum-free DMEM, supplemented with 10 μg/mL of DEAE dextran (Sigma #D9885). After 48 h, transduced cells were selected by puromycin treatment for 20 days (1 μg/mL; Sigma #P8833).

**CRISPR/Cas9 screen**. In total, $4 \times 10^7$ A549-ACE2 cells were treated with IFNα (200 U/mL). 16 h later, cells were infected at an MOI of 1 in serum-free media complemented with TPCK-trypsin and IFNα (200 U/mL). After 90 min, the viral inoculum was removed, and cells were maintained in DMEM containing 5% FBS and IFNα (200 U/mL). After 24 h, cells were harvested and fixed for 15 min in Formalin 1%. Fixed cells were washed in cold FACS buffer containing PBS, 2% Bovine Serum Albumin (Sigma-Aldrich #A2153-100G), 2 mM EDTA (Invitrogen #15575-038) and 0.1% Saponin (Sigma-Aldrich #S7900-100G). Cells were incubated for 30 min at 4 °C under rotation with primary antibody diluted in FACS buffer. Incubation with the secondary antibody was performed during 30 min at 4 °C under rotation. Stained cells were resuspended in cold sorting buffer containing PBS, 2% FBS, 25 mM Hepes (Sigma-Aldrich #H0887-100ML) and 5 mM EDTA. Infected cells were sorted on a BD FACS Aria Fusion. Sorted and control (non-infected, not IFN-treated) cells were centrifuged (20 min, 2,000 g) and resuspended in lysis buffer (NaCl 300 mM, SDS 0.1%, EDTA 10 mM, EGTA 20 mM, Tris 10 mM) supplemented with 1% Proteinase K (Qiagen #19133) and 1% RNAse A/T1 (ThermoFisher #EN0551) and incubated overnight at 65 °C. Two consecutive phenol-chloroform (Sigma #P3803-100ML) extractions were performed and DNA was recovered by ethanol precipitation. Nested PCR was performed using the Herculase II Fusion DNA Polymerase (Agilent, #600679) and the DNA oligos indicated in Supplementary Table 1. PCR1 products were purified using QIAquick PCR Purification kit (Qiagen #28104). PCR2 products were purified using Agencourt AMPure XP Beads (Beckman Coulter Life Sciences #A63880). DNA concentration was determined using Qubit dsDNA HS Assay Kit (Thermo Fisher #Q32854) and adjusted to 2 nM prior to sequencing. NGS was performed using the NextSeq 500/550 High Output Kit v2.5 75 cycles (Illumina #20024906).

**Screen analysis**. Reads were demultiplexed using bcl2fastq Conversion Software v2.20 (Illumina) and fastx_toolkit v0.0.13. Sequencing adapters were removed using cutadapt v1.9.1.[70] The reference library was built using bowtie2 v2.2.9.[71] Read mapping was performed with bowtie2 allowing 1 seed mismatch in -local mode and samtools v1.9.[72] Mapping analysis and gene selection were performed using MAGeCK v0.5.6, normalizing the data with default parameters. sgRNA and gene enrichment analyses are available in Supplementary Data 1 and 2, respectively and full MAGeCK output at https://github.com/Simon-LoriereLab/crispr_isg_sarscov2.

**Generation of multi-guide gene knockout cells**. 3 sgRNAs per gene were designed (Supplementary Table 2). 10 pmol of NLS-Sp.Cas9-NLS (SpCas9) nuclease (Aldevron #9212) was combined with 30 pmol total synthetic sgRNA (10 pmol for each sgRNA) (Synthego) to form RNPs in 20 μL total volume with SE Buffer (Lonza #V5SC-1002). The reaction was incubated at room temperature for 10 min. In total, $2 \times 10^5$ cells per condition were pelleted by centrifugation at $100 \times g$ for 3 min, resuspended in SE buffer and diluted to $2 \times 10^4$ cells/μL. 5 μL of cell solution was added to the pre-formed RNP solution and gently mixed. Nucleofections were performed on a Lonza HT 384-well nucleofector system (Lonza #AAU-1001) using program CM-120. Immediately following nucleofection, each reaction was transferred to a 96-well plate containing 200 μL of DMEM 10% FBS ($5 \times 10^4$ cells per well). Two days post-nucleofection, DNA was extracted using DNA QuickExtract (Lucigen #QE09050). Cells were lysed in 50 μL of QuickExtract solution and incubated at 68 °C for 15 min followed by 95 °C for 10 min. Amplicons were generated by PCR amplification using NEBNext polymerase (NEB #M0541) or AmpliTaq Gold 360 polymerase (ThermoFisher #4398881) and the primers indicated in Supplementary Table 3. PCR products were cleaned-up and analyzed by Sanger sequencing. Sanger data files and sgRNA target sequences were input into Inference of CRISPR Edits (ICE) analysis https://ice.synthego.com/#/ to determine editing efficiency and to quantify generated indels.[73] Percentage of alleles edited is shown in Table 1 ($n = 3$).

**Hit validation**. In total, $2.5 \times 10^4$ A549-ACE2 KO cells were seeded in 96-well plates 18 h before the experiment. Cells were treated with IFNα and infected as described above. At 72 h post-infection, supernatants and cellular monolayers were harvested in order to perform qRT-PCR and plaque assay titration. Infectious

supernatants were heat-inactivated at 80 °C for 10 min. For intracellular RNA, cells were lysed in a mixture of Trizol Reagent (Invitrogen #15596018) and PBS at a ratio of 3:1. Total RNA was extracted using the Direct-zol 96 RNA kit (Zymo Research #R2056) or the Direct-zol RNA Miniprep kit (Zymo Research #R2050). For SARS-CoV-2 detection, qRT-PCR was performed either directly on the inactivated supernatants or on extracted RNA using the Luna Universal One-Step RT-qPCR Kit (NEB #E3005E) in a QuantStudio 6 thermocycler (Applied Biosystems) or in a StepOne Plus thermocycler (Applied Biosystems). The primers used are described in Supplementary Table 4. Cycling conditions were the following: 10 min at 55 °C, 1 min at 95 °C and 40 cycles of 95 °C for 10 s and 60 °C for 1 min. Results are expressed as genome copies/mL as the standard curve was performed by diluting a commercially available synthetic RNA with a known concentration (EURM-019, JRC). For SARS-CoV and MERS-CoV, qRT-PCR were performed using FAM-labelled probes (Eurogentec) and the Superscript III Platinum One-Step qRT-PCR System (Thermo Fisher Scientific, #11732020). The cycling conditions were the following: 20 min at 55 °C, 3 min at 95 °C and 50 cycles of 95 °C for 15 s and 58 °C for 30 s. The primers used are described in Supplementary Table 4. Standard curves were performed using serial dilutions of RNA extracted from and SARS-CoV and MERS-CoV viral culture supernatants of known infectious titer. For plaque assay titration, VeroE6 cells were seeded in 24-well plates ($10^5$ cells per well) and infected with serial dilutions of infectious supernatant diluted in DMEM during 1 h at 37 °C. After infection, 0.1% agarose semi-solid overlays were added. At 72 h post-infection, cells were fixed with Formalin 4% (Sigma #HT501128-4L) and plaques were visualized using crystal violet coloration. Time-course experiments were performed the same way, except that supernatants and cellular monolayers were harvested at 0 h, 2 h, 4 h, 6 h, 8 h, 10 h and 24 h post-infection.

**SARS-CoV-2, SARS-CoV and MERS-CoV infection assays**. A549-ACE2 cells were infected by incubating the virus for 1 h with the cells maintained in DMEM supplemented with 1 μg/ml TPCK-trypsin (Sigma #4370285). The viral input was then removed and cells were kept in DMEM supplemented with 2% FBS. For 293T-ACE2 cells, infections were performed without TPCK-trypsin. MERS-CoV and SARS-CoV infections were performed in DMEM supplemented with 2% FBS and cells were incubated 1 h at 37 °C 5% CO2. Viral inoculum was then removed and replaced by fresh DMEM supplemented with 2% FBS. All experiments involving infectious material were performed in Biosafety Level 3 facilities in compliance with Institut Pasteur's guidelines and procedures. When indicated, SARS-CoV-2 infected cells were stained for intracellular Spike levels as described below.

**Yellow fever virus and measles virus infection assays**. Cells were infected with YFV (at an MOI of 0.3) or MeV-GFP (at an MOI of 0.2) in DMEM without FBS for 2 h in small volume of medium to enhance contacts with the inoculum and the cells. After 2 h, the viral inoculum was replaced with fresh DMEM 10% FBS 1% P/S. FACS analysis were performed at 24 h p.i. Cells were fixed and permeabilized using BD Cytofix/Cytoperm (Fisher Scientific, # 15747847) for 30 min on ice (all the following steps were performed on ice and centrifuged at 4 °C) and then washed tree times with wash buffer. Cells infected with YFV were incubated with the pan-flavivirus anti-Env 4G2 antibody for 1 h at 4 °C and then with Alexa 488 anti-mouse IgG secondary antibodies (Thermo Fisher, #A28175) for 45 min at 4 °C in the dark. Non-infected, antibody-stained samples served as controls for signal background. The number of cells infected with MeV-GFP were assessed with the GFP signal, using non-infected cells as controls. Data were acquired with an Attune NxT Acoustic Focusing Cytometer (Life technologies) and analyzed using FlowJo software.

**Entry assays**. Cells were seeded at $1 \times 10^5$ cells per well in 24-well plates in DMEM with 1% FBS. The next day, cells were infected with VSV* (MOI 0.001) or VSV*ΔG-S (MOI 7) in DMEM without FBS. The virus suspension was removed after 2 h and replaced with DMEM with 1% FBS. 16 h p.i., cells were washed once with PBS, trypsinized and subsequently fixed in 4% PFA. Fixed cells were washed once with PBS and analyzed by flow cytometry. The percentage of infected cells was identified based on GFP expression.

**Overexpression assay**. In total, $2 \times 10^5$ 293T-ACE2 cells were seeded in a 24-well plate 18 h before the experiment. Cells were transfected with 500 ng of plasmids expressing HA-DAXX WT, HA-DAXX 15KR and HA-DAXXΔSIM plasmids, using Fugene 6 (Promega # E2691), following the manufacturer's instructions. HA-NBR1 was used as negative control. After 24 h cells were infected at the indicated MOI in DMEM 2% FBS. When indicated, cells were treated with 10 mM of remdesivir (MedChemExpress #HY-104077) at the time of infection. For flow cytometry analysis, cells were fixed with 4% formaldehyde and permeabilized in a PBS 1% BSA 0.025% saponin solution for 30 min prior to staining with corresponding antibodies for 1 h at 4 °C diluted in the permeabilization solution. Samples were acquired on a BD LSR Fortessa and analyzed using FlowJo. Total RNA was extracted using a RNeasy Mini kit and submitted to DNase treatment (Qiagen). RNA concentration and purity were evaluated by spectrophotometry (NanoDrop 2000c, ThermoFisher). In addition, 500 ng of RNA were reverse transcribed with both oligo dT and random primers, using a PrimeScript RT

Reagent Kit (Takara Bio) in a 10 mL reaction. Real-time PCR reactions were performed in duplicate using Takyon ROX SYBR MasterMix blue dTTP (Eurogentec) on an Applied Biosystems QuantStudio 5 (ThermoFisher). Transcript levels were quantified using the following program: 3 min at 95 °C followed by 35 cycles of 15 s at 95 °C, 20 s at 60 °C, and 20 s at 72 °C. Values for each transcript were normalized to expression levels of *RPL13A*. The primers used are indicated in Supplementary Table 4.

**Microscopy immunolabeling and imaging**. 293T-ACE2 cells were cultured and infected with SARS-CoV-2 as described above. When indicated, cells were treated with 50 μM of GRL0617 (MedChemExpress #HY-117043) at the time of infection. Cultures were rinsed with PBS and fixed with 4% paraformaldehyde (electronic microscopy grade; Alfa Aesar) in PBS for 10 min at room temperature, treated with 50 mM NH4Cl for 10 min, permeabilized with 0.5% Triton X-100 for 15 min, and blocked with 0.3% BSA for 10 min. Cells were incubated with primary and secondary antibodies for 1 h and 30 min, respectively, in a moist chamber. Nuclei were labeled with Hoechst dye (Molecular Probes). Images were acquired using a LSM700 (Zeiss) confocal microscope equipped with a 63X objective or by Airyscan LSM800 (Zeiss). Image analysis and quantification was performed using ImageJ (Fiji) v2.1.

**Western blot**. 293T-ACE2 cells were transfected with the indicated plasmids or treated with the indicated concentrations of GRL0617; with 10 μM of Masitinib (MedChemExpress #HY-10209); or with 10 μM of MG132 (SIGMA #M7449), an inhibitor of the proteasome and infected with SARS-CoV-2. Cell lysates were prepared using RIPA lysis and extraction buffer (ThermoFisher #89901). Protein concentration was determined using Bradford quantification. Proteins were denatured using 4X Bolt LDS Sample Buffer (Invitrogen) and 10X Bolt Sample Reducing Agent (Invitrogen). 40 μg of proteins were denatured and loaded on 12% ProSieve gel and then subjected to electrophoresis. Gels were then transferred (1 h, 90 V) to Western blotting membranes, nitrocellulose (GE Healthcare #GE10600002) using Mini Trans-Blot Electrophoretic Transfer Cell (Biorad #1703930EDU). Membranes were blocked with 5% BSA in PBS (blocking buffer) and incubated with primary antibodies diluted in blocking buffer. Membranes were washed and incubated with secondary antibodies diluted in blocking buffer. Chemiluminescent acquisitions were performed on a Chemidoc™ MP Imager and analyzed using Image Lab v6 software (Bio-Rad Laboratories).

**Flow cytometry**. For flow cytometry analysis, all cells were fixed with 4% formaldehyde. For intracellular staining, cells were permeabilized in a PBS 1% BSA 0.025% saponin solution for 30 min prior to staining with corresponding primary antibodies for 1 h at 4 °C and then secondary antibodies for 45 min at 4 °C, diluted in the permeabilization solution. Acquisition was done with BD Fortessa and Attune NxT cytometers. Data was analyzed with the FlowJo software (Treestar Inc., Oregon, USA v10.8.1).

**Single-cell RNAseq analysis**. Single cell RNAseq analysis were performed in the BioTuring Browser software v2.8.42 developed by BioTuring, using a dataset made available by Liao et al.[43] (GSE145926). All processing steps were done by BioTuring Browser.[74] Cells with less than 200 genes and mitochondrial genes higher than 10% were excluded from the analysis.

**Statistical analysis**. Graphpad Prism v9.1.0 was used for statistical analyses. Linear models were computed using Rstudio v1.2.1335.

**Reporting summary**. Further information on research design is available in the Nature Research Reporting Summary linked to this article.

## Data availability
CRISPR/Cas9 screen NGS raw (fastq) and preprocessed data (read counts) is available in NCBI GEO (GSE173418); sgRNA and gene enrichment analyses are available in Supplementary Data 1 and 2, respectively and full MAGeCK output at https://github.com/Simon-LoriereLab/crispr_isg_sarscov2. The single cell data from Liao et al. 2020 Nature is available in NCBI GEO (GSE145926). Source data are provided with this paper.

## Code availability
All code used in this study is available at https://github.com/Simon-LoriereLab/crispr_isg_sarscov2.

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

## Acknowledgements

We thank the Cytometry Platform, Center for Technological Resources and Research, Institut Pasteur, for cell sorting experiments. This work was funded by the Institut Pasteur Coronavirus Task Force, CNRS (UMR 3569), the Labex IBEID (ANR-10-LABX-62-IBEID), the Occitanie region and by the ANR (ANR-20-COVI-000, projects IDIS-COVR to M.V. and Alpha-COV to S.N.). A.M.K. is supported by a grant of the French Ministry of Higher Education, Research and Innovation. G.M. is supported by a grant from the Agence nationale de recherches sur le sida et les hépatites virales (ANRS). S.M.A. is supported by the Pasteur-Paris University (PPU) International PhD Program. We thank Michael Emerman, Daniel Marc and Ignacio Caballero-Posadas for helpful comments on the manuscript. Figure 1 was created with BioRender.com.

## Author contributions

F.R. designed the research project. F.R., N.J., S.N. and M.V. secured the funding for the study. A.M.K., S.M.A., A.H., N.A., S.N., G.M., D.Q.T., M.C., T.V. and F.R. performed and analyzed the in vitro experiments. F.P. produced the stocks of lentiviruses. J.C.S., J.O. and K.H. generated and validated KO cell lines. T.B. performed the single-cell RNAseq data analysis. S.M. and F.D. performed the SARS-CoV and MERS-CoV experiments. A.B. and E.S.L. performed the bio-informatic analyses of the CRISPR/Cas9 screen. G.Z. supplied the VSV* and VSV*ΔG-S viruses used for entry assays and share protocols. M.O., T.B., O.S., N.J., S.N., S.V.D.W. and M.V. analyzed the data and supervised the project. A.M.K., N.J., S.N. and F.R. wrote the manuscript. All authors edited the manuscript. Correspondence and requests for materials should be addressed to either M.V., N.J., S.N. or F.R.

## Competing interests

J.C.S., J.O. and K.H. are employees and shareholders from Synthego Corporation. The remaining authors declare no competing interests.
