## [Peer Review File · Nature Communications]

Reviewers' Comments:

Reviewer #1:

Remarks to the Author:

In their manuscript, Mac Kain et al. describe their CRISPR/Cas9 screen for genes that can inhibit SARS-CoV-2 infection. The screen results confirm previously identified pro- and antiviral host genes but also identify DAXX as a new restriction factor of SARS-CoV-2. The authors show that DAXX relocalizes during SARS-CoV-2 infection and they claim to show that SARS-CoV-2 counteracts restriction by DAXX. The screen adds an important new piece of information to our current understanding of innate immune control of SARS-CoV-2. However, the manuscript does not provide any insights into the mechanism of antiviral activity and in my opinion, the claim that SARS-CoV-2 counteracts DAXX restriction is not supported by the data provided. Thus, the report seems preliminary to me.

Major points:

- The authors should provide information on the mechanism of antiviral activity of DAXX. The authors claim to provide evidence that DAXX targets viral transcription but this conclusion is not supported by the data provided (see next point). Furthermore, the authors show that two DAXX mutants also possess antiviral activity and that DAXX relocalizes upon infection but these two points do not provide much insight into the antiviral mechanism. Using their FACS-based assay (fig. 3) the authors could generate a panel of additional DAXX constructs to map the domains required for antiviral activity. Moreover, the authors should test if DAXX can interfere with viral entry, e.g. in a VLP assay, and test for viral transcript and protein levels at early time points of a high MOI infection to map the block to a certain stage of the viral replication cycle.
- I do not agree with the conclusions drawn from fig. 2e: The authors infect cells with SARS-CoV-2 at MOI 0.1 and measure viral transcript levels at 72h post infection. In this setting, it is highly likely that multiple rounds of viral replication have occurred and thus measuring viral transcript levels is just another way of measuring virus propagation. These results do not allow for the conclusion that DAXX likely targets viral transcription. If the authors wanted to test which step of the viral replication cycle is targeted by DAXX a high MOI infection should be done and time points such as 4, 6, 8, 10h post infection should be analyzed for viral transcript levels, viral protein expression etc..
- The authors claim that SARS-CoV-2 counteracts restriction by DAXX and they furthermore hypothesize that the protease PLpro could be responsible for DAXX degradation. However, the data provided are preliminary and in my opinion do not allow for the conclusions drawn. The authors should test protein levels for a number of other proteins to see if there is a specific loss of DAXX or if there is a more general host cell shut down, which leads to reduced levels of many proteins. GAPDH as the only control in the western blot is not sufficient. Furthermore, the observation that addition of GRL-0617 rescues DAXX levels to a certain extent should be investigated more thoroughly. I cannot see the specific rescue of DAXX levels in infected cells that the authors claim to show in fig. 5b. Thus, I would conclude that the inhibitor just blocks viral replication, which could then prevent host cell shut off. At a minimum the authors should include specific 3CLpro inhibitors, as well as remdesivir in addition to the PLpro inhibitor and include stainings for viral proteins in the western blot. Data on cell viability upon inhibitor treatment would be helpful and a quantitative read-out for virus inhibition by GRL-0617 would also be required.

Minor points:

- In my opinion, figure 2c is not very informative the way it is presented. I would use the data on infectious titers (2b) rather than viral RNA (2a) in the supernatant as infectious virus is the more important measurement. I would also consider using a linear scale to illustrate differences better. In the current version, the bars look more or less the same except for the significance ratings.
- In figure 2d, I would change "historical strain" to "wave 1 isolate" or use the strain name. "Historical strain" is not informative.

Reviewer #2:

Remarks to the Author:

The manuscript by Kain et al. uses a CRISPR subpool screen (~2000 genes) to identify host genes which inhibit SARS-CoV-2 protein expression. The screen utilized a CRISPRa subpool the authors

previously described and probed for antiviral genes in the presence of IFN α , thus enhancing the chance that they would find antiviral ISGs. In addition to previously validated Ly6E, they found the protein DAXX to be antiviral. Interestingly DAXX was not induced by IFN in the cells used nevertheless it had antiviral activity. The authors validated DAXX and partially unveiled the mechanism by which DAXX inhibits SARS-CoV-2 infection. While there are novel findings here, the study is lacking in key mechanistic details which dampen enthusiasm.

Major

1. What stage of the viral life cycle is counteracted by DAXX? At a minimum the authors should determine whether this is viral entry or post-entry which could be done using a variety of assays (pseudovirus, transfection of vRNA into KO cells, etc)
2. The authors should provide additional insight into the mechanism of virus-induced DAXX degradation. What viral protein degrades DAXX? Transfection of key viral ORFs which have previously been described should be performed to evaluate whether any one viral gene product can degrade DAXX. This is a straightforward and informative experiment. It should also be assessed whether DAXX downregulation happens at the transcript or protein level? Does it require the proteasome?
3. IFN itself can inhibit cell replication and induce cell death. Therefore it is not ideal that the authors compared virus+IFN to no virus/no IFN. The authors should compare virus+IFN to no virus+IFN.
4. What is the specificity of DAXX for SARS-CoV-2? Testing the antiviral activity of DAXX against other coronaviruses (or RNA viruses) would provide additional insight into the mechanism and breadth of DAXX function.

Minor revisions

5. -DAXX is known to regulate apoptosis. It would be helpful to know if cells with altered levels of DAXX show differences in cell death.
6. For table 1, was this N=1? What is the variance of editing amongst repeats?
7. Microscopy images should be quantified in Figure 3.
8. -In figure 2, the authors assess antiviral activity of "ISGs" but how many of these genes are actually ISGs in the A549s tested? DAXX is not an ISG in A549s so this should be clarified.
9. The term "PFU equivalents" should be changed to "viral RNA" or "viral genomes" The PFU to RNA ratio can vary wildly (by more than 6-logs) across viral stocks, cell types, etc so this is not appropriate.
10. -For reader clarity, it would be preferable to relabel viruses in Figure 2d with more commonly known names i.e. B.1.1.7. Also, "historical strain" would be better as "reference virus" or "prototype virus" (or just give the actual name of the virus).
11. -The label of the Y axis on Figure 2e is confusing. Does a value of 2 mean 100x more viral transcript than GAPDH mRNA or does it mean 100x more viral transcript than the NTC?
12. - In figure 1b, what is the -log₁₀ label on the Y-axis referring to (FDR? P-value?)?
13. -In figure 1c, what does the Log₂ FC refer to exactly? Guide enrichment of the virus+IFN condition relative to the no virus and no IFN condition? Is it the spike+ vs the spike- cells?
14. Line 31: MDA5 needs to be in all caps.

Reviewer #3:

Remarks to the Author:

Using a wider ISG CRISPR sgRNA library, Kain et al. identified genes, specifically the DAXX, as restriction factors for SARS-CoV-2 infection. DAXX is a known antiviral host factor against DNA viruses, but its function is unrelated to the SUMOylation pathway. In addition, the authors showed that the activity of DAXX on SARS-CoV-2 is IFN-independent, and the protein is degraded upon virus infection. Generally, the finding of DAXX as antiviral host factor for SARS-CoV-2 is novel, but the mechanisms of action remains to be further explored.

One life cycle of coronavirus infection is around 8-12h, thus, the reduced viral transcript measured at 72h post-infection as shown in Fig.2e and probably in Fig. S4c (not sure about the time point) is too late to draw the conclusion that DAXX targets viral transcription (line 148). Although the results indicated that DAXX reduced the virus production, viral transcript, re-located to the

cytoplasm and co-stained with the viral dsRNA, more data is required to support the notion that DAXX affects the virus replication or transcription, or probably affects the early entry step. RNA replicon assay, genomic RNA transfection to bypass the entry, or virion binding/internalization assay may be required to demonstrate.

The phenomenon that DAXX could be degraded upon virus infection is not new, as adenovirus and FMDV have similar feature. It is of significance to further figure out how SARS-CoV-2 infection decreases the DAXX protein level or probably the RNA level.

We kindly thank all three reviewers for assessing our work and for suggesting experiments that increase the quality of this manuscript. Our answers are indicated below in blue font.

REVIEWER COMMENTS

Reviewer #1 (Remarks to the Author):

In their manuscript, Mac Kain et al. describe their CRISPR/Cas9 screen for genes that can inhibit SARS-CoV-2 infection. The screen results confirm previously identified pro- and antiviral host genes but also identify DAXX as a new restriction factor of SARS-CoV-2. The authors show that DAXX relocalizes during SARS-CoV-2 infection and they claim to show that SARS-CoV-2 counteracts restriction by DAXX. The screen adds an important new piece of information to our current understanding of innate immune control of SARS-CoV-2. However, the manuscript does not provide any insights into the mechanism of antiviral activity and in my opinion, the claim that SARS-CoV-2 counteracts DAXX restriction is not supported by the data provided. Thus, the report seems preliminary to me.

We thank reviewer #1 for his/her interest in our work. In the revised version of our manuscript, we provide additional insights into the mechanism of DAXX antiviral activity. Moreover, we have now identified the SARS CoV-2 papain-like protease (PLpro) as the factor responsible for DAXX degradation.

Major points:

- The authors should provide information on the mechanism of antiviral activity of DAXX. The authors claim to provide evidence that DAXX targets viral transcription but this conclusion is not supported by the data provided (see next point).

We performed additional experiments that are demonstrating that DAXX acts at the step of viral transcription (see our detailed answer below).

Furthermore, the authors show that two DAXX mutants also possess antiviral activity and that DAXX relocalizes upon infection but these two points do not provide much insight into the antiviral mechanism. Using their FACS-based assay (fig. 3) the authors could generate a panel of additional DAXX constructs to map the domains required for antiviral activity.

Our data demonstrates that DAXX acts independently of its SIM domain, which, we believe, is an important aspect of its mechanism. Indeed, most of the previously published functions of DAXX are linked to the recruitment of SUMOylated partners through this domain (Shi *et al.* 2007 and Lin *et al.* 2006, cited in our manuscript), although some functions such as interaction with ATRX or the recently described role of DAXX as a protein chaperone are also SIM-independent. Thus, the fact that the antiviral mechanism of DAXX is SIM-independent is a novel aspect worthy to be published. This is now emphasized in the text:

“ Some DAXX functions, such as interaction with the chromatin remodeler ATRX (29) or its recently described role as a protein chaperone (58) are, however, SIM-independent. Future work should determine which DAXX domains and residues are required for its antiviral activity. “

We consider that an in depth-characterization of the domains and residues of DAXX required for antiviral activity is out of the scope of this manuscript. However, it represents an interesting follow-up prospect for future research.

Moreover, the authors should test if DAXX can interfere with viral entry, e.g. in a VLP assay, and test for viral transcript and protein levels at early time points of a high MOI infection to map the block to a certain stage of the viral replication cycle.

I do not agree with the conclusions drawn from fig. 2e: The authors infect cells with SARS-CoV-2 at MOI 0.1 and measure viral transcript levels at 72h post infection. In this setting, it is highly likely that multiple rounds of viral replication have occurred and thus measuring viral transcript levels is just another way of measuring virus propagation. These results do not allow for the conclusion that DAXX likely targets viral transcription. If the authors wanted to test which step of the viral replication cycle is targeted by DAXX a high MOI infection should be done and time points such as 4, 6, 8, 10h post infection should be analyzed for viral transcript levels, viral protein expression etc.

We agree with the reviewer that measuring viral RNA yield at 72h p.i. was not sufficient to claim that DAXX restricts SARS-CoV-2 at the step of viral transcription. This claim was also based on data shown in former Fig. S4 (now Fig. S5), in which we assessed the abundance of viral transcripts at 24h p.i. in DAXX overexpressing cells. To further demonstrate that DAXX acts at the step of viral transcription, we performed the experiment suggested by the reviewer, which is now shown in the new Fig. 3. These novel results clearly show a significant increase in viral transcript levels starting between 8h and 10h p.i. in DAXX KO cells as compared to control cells. We did not observe an effect of DAXX at earlier time points, suggesting that DAXX acts at the step of transcription rather than on viral entry, further validating our previous results.

Figure 3: DAXX restricts SARS-CoV-2 transcription. A549-ACE2 WT or DAXX KO cells were infected at a MOI 1 in triplicate. Cell monolayers were harvested at the indicated time points, and total RNA was extracted. The levels of viral RNA (a: 5' UTR; b: RdRp) were determined by RT-qPCR and normalized against GAPDH levels. The mean of 3 independent experiments is represented. Statistics: Dunnett's test on a linear model, * p-value < 0.05, ** p-value < 0.01, *** p-value < 0.001.

The authors claim that SARS-CoV-2 counteracts restriction by DAXX and they furthermore hypothesize that the protease PLpro could be responsible for DAXX degradation. However, the data provided are preliminary and in my opinion do not allow for the conclusions drawn.

The authors should test protein levels for a number of other proteins to see if there is a specific loss of DAXX or if there is a more general host cell shut down, which leads to reduced levels of many proteins. GAPDH as the only control in the western blot is not sufficient.

We now provide an updated version of our Western Blot analysis, shown in **Fig. 6a**, in which we assessed the level of expression of the following cellular proteins: Lamin B, Hsp90, actin, tubulin, TRIM22 and RIG-I. We also included in this novel analysis the quantification of the level of expression of the viral protein S. None of the cellular proteins tested, apart from DAXX, exhibited reduced levels in SARS-CoV-2 infected cells as compared to non-infected cells, suggesting that the reduction of DAXX levels is due to a specific degradation rather than to global host cell shut down.

Figure 6: SARS-CoV-2 PLpro induces the proteasomal degradation of DAXX. a: DAXX degradation upon infection. 293T-ACE2 cells were infected with SARS-CoV-2 at a MOI 0.1. After 24h, cells were harvested and levels of DAXX, Lamin B, HSP90, Actin, GAPDH, Tubulin, TRIM22, RIG-I and of the viral protein spike were analyzed by Western Blot.

Furthermore, the observation that addition of GRL-0617 rescues DAXX levels to a certain extent should be investigated more thoroughly. I cannot see the specific rescue of DAXX levels in infected cells that the authors claim to show in fig. 5b. Thus, I would conclude that the inhibitor just blocks viral replication, which could then prevent host cell shut off. At a minimum the authors should include specific 3CLpro inhibitors, as well as remdesivir in addition to the PLpro inhibitor and include stainings for viral proteins in the western blot. Data on cell viability upon inhibitor treatment would be helpful and a quantitative read-out for virus inhibition by GRL-0617 would also be required.

We thank the reviewer for these suggestions. We have now investigated more thoroughly the mechanisms by which SARS-CoV-2 induced the degradation of DAXX in the novel **Fig. 6**.

Figure 6: SARS-CoV-2 PLpro induces the proteasomal degradation of DAXX. **a: DAXX degradation after infection.** 293T-ACE2 cells were infected with SARS-CoV-2 at MOI 0.1. After 24h, cells were harvested and levels of DAXX, Lamin B, HSP90, Actin, GAPDH, Tubulin, TRIM22, RIG-I and of the viral protein spike were analyzed by Western Blot. **b: GRL0617 and MG132 treatment restores DAXX expression.** 293T-ACE2 cells were infected with SARS-CoV-2 at MOI 0.1. When indicated, cells were pretreated 2h before infection with GRL0617 (at the indicated concentrations), or with MG132 (10 μ M), a proteasome inhibitor, or Masitinib (10 μ M) a 3CL inhibitor. After 24h, cells were harvested and levels of DAXX, GAPDH and of the viral protein spike were analyzed by Western Blot. **c: GRL0617 treatment restores DAXX localization.** 293T-ACE2 cells were infected with SARS-CoV-2 at MOI 0.1. 24h post-infection, cells were labelled with Hoescht and with antibodies against dsRNA (detecting viral RNA, in green) and HA (detecting DAXX, in red). When indicated, cells were treated with 50 μ M of GRL0617 at the time of infection. Scale bars correspond to 10 μ m. **d-f: Nsp3 induces DAXX degradation.** **D:** 293T-ACE2 cells were transfected with 1 μ g of the indicated viral proteins. After 24h, the levels of DAXX and GAPDH were analyzed by Western Blot. **E:** 293T-ACE2 cells were transfected with the indicated amounts of Nsp3. After 24h, the levels of DAXX and GAPDH were analyzed by Western Blot. **f:** 293T-ACE2 cells were transfected with 1 μ g of Nsp3. 6 hours post transfection, cells were also, when indicated, treated with 50 μ M of GRL0617. 24h after transfection, the levels of DAXX and GAPDH were analyzed by Western Blot.

In **Fig. 6b**, we now show that treating cells with GRL0617, an inhibitor of PLpro (here used at two different concentrations), but not with masitinib, the 3CLpro inhibitor, prevents SARS-CoV-2-mediated degradation of DAXX. In addition, we have directly tested the effect of several viral proteins on DAXX degradation, and found that overexpression of Nsp3, which encodes PLpro, leads to DAXX degradation in a dose dependent fashion (**Fig. 6d-e**), a process that was inhibited by GRL0617 (**Fig. 6f**). Moreover, we found that treatment with MG132, a well-characterized inhibitor of the proteasome, also prevented DAXX degradation (**Fig. 6b**). We are also showing that SARS-CoV-2 infection had no effect on DAXX mRNA levels (**Fig. S6**). Taken together, these novel results indicate that PLpro degrades DAXX in a proteasome-dependent mechanism.

As suggested by the reviewer, we have also included viability data following inhibitor treatment (**Fig. S7**) and staining for SARS-CoV-2 spike (**Fig. 6b**). While the inhibitors had no detectable effect on viability (as assessed by FSC/SSC profiles), they did, as expected, decrease the synthesis of viral proteins. The fact that GRL0617 prevents DAXX degradation

induced by Nsp3 overexpression (**Fig. 6e**) is however a strong argument in favor of a direct effect of GRL0617 on DAXX, rather than on host shut-off.

Supplementary Figure 7: FACS analyses of 293T-ACE2 treated with inhibitors.

The size (FSC) and granularity (SSC) of the cells used for the western-blots shown in **Fig. 6b** were evaluated by flow cytometry. The estimated percentage of live cells is indicated.

Minor points:

- In my opinion, figure 2c is not very informative the way it is presented. I would use the data on infectious titers (2b) rather than viral RNA (2a) in the supernatant as infectious virus is the more important measurement. I would also consider using a linear scale to illustrate differences better. In the current version, the bars look more or less the same except for the significance ratings.

We used “viral RNA” and not “infectious titers” to express the results since some infectious titers values were below the limit of detection of the assays. The results were already presented in a linear scale, but on log transformed titers (in order to take into account variability between experiments). We have changed the scale of the axis in order to make it easier to read.

- In figure 2d, I would change “historical strain” to “wave 1 isolate” or use the strain name. “Historical strain” is not informative.

We have changed the names of all strains to follow the recently published WHO guidelines, as follows:

- *BetaCoV/France/IDF0372/2020 (historical): Lineage B*
- *hCoV-19/France/IDF-IPP11324/2020 (or UK): Lineage B.1.1.7 (Alpha)*
- *hCoV-19/France/PDL-IPP01065/2021 (20H/501Y.V2 or SA): Lineage B.1.351 (Beta)*
- *hCoV-19/Japan/TY7-501/2021 (20J/501Y.V3 or Brazil): Lineage P1 (Gamma)*

Reviewer #2 (Remarks to the Author):

The manuscript by Kain et al. uses a CRISPR subpool screen (~2000 genes) to identify host genes which inhibit SARS-CoV-2 protein expression. The screen utilized a CRISPRa subpool the authors previously described and probed for antiviral genes in the presence of IFN α , thus enhancing the chance that they would find antiviral ISGs. In addition to previously validated Ly6E, they found the protein DAXX to be antiviral. Interestingly DAXX was not induced by IFN in the cells used nevertheless it had antiviral activity. The authors validated DAXX and partially unveiled the mechanism by which DAXX inhibits SARS-CoV-2 infection. While there are novel findings here, the study is lacking in key mechanistic details which dampen enthusiasm.

We thank the reviewer for his/her kind comments. To further characterize the mechanism of action of DAXX, we have performed additional experiments, as detailed below.

Major revisions

1. What stage of the viral life cycle is counteracted by DAXX? At a minimum the authors should determine whether this is viral entry or post-entry which could be done using a variety of assays (pseudovirus, transfection of vRNA into KO cells, etc)

To identify which stage of the viral cycle is counteracted by DAXX, we performed high MOI infection and assessed viral RNA levels at multiple time points in WT and DAXX KO cells (novel **Fig. 3**). These novel results clearly show that an increase in viral transcript levels around 8-10h p.i. in DAXX KO cells, as compared to control cells. We did not observe an effect of DAXX knock-out on viral replication at earlier time points, arguing that DAXX acts at the step of transcription rather than on viral entry, further validating previous results obtained in the context of DAXX over-expression.

Figure 3: DAXX restricts SARS-CoV-2 transcription. A549-ACE2 WT or DAXX KO cells were infected at a MOI 1 in triplicate. Cell monolayers were harvested at the indicated time points, and total RNA was extracted. The levels of viral RNA (a: 5' UTR; b: RdRp) were determined by RT-qPCR and normalized against GAPDH levels. The mean of 3 independent experiments is represented. Statistics: Dunnett's test on a linear model, * p-value < 0.05, ** p-value < 0.01, *** p-value < 0.001.

2. The authors should provide additional insight into the mechanism of virus-induced DAXX degradation. What viral protein degrades DAXX? Transfection of key viral ORFs which have previously been described should be performed to evaluate whether any one viral gene product can degrade DAXX. This is a straightforward and informative experiment. It should also be assessed whether DAXX downregulation happens at the transcript or protein level? Does it require the proteasome?

We thank the reviewer for suggesting these additional mechanistic experiments, now shown in the novel **Fig. 6**.

Figure 6: SARS-CoV-2 PLpro induces the proteasomal degradation of DAXX. **a: DAXX degradation after infection.** 293T-ACE2 cells were infected with SARS-CoV-2 at MOI 0.1. After 24h, cells were harvested and levels of DAXX, Lamin B, HSP90, Actin, GAPDH, Tubulin, TRIM22, RIG-I and of the viral protein spike were analyzed by Western Blot. **b: GRL0617 and MG132 treatment restores DAXX expression.** 293T-ACE2 cells were infected with SARS-CoV-2 at MOI 0.1. When indicated, cells were pretreated 2h before infection with GRL0617 (at the indicated concentrations), or with MG132 (10 μ M), a proteasome inhibitor, or Masitinib (10 μ M) a 3CL inhibitor. After 24h, cells were harvested and levels of DAXX, GAPDH and of the viral protein spike were analyzed by Western Blot. **c: GRL0617 treatment restores DAXX localization.** 293T-ACE2 cells were infected with SARS-CoV-2 at MOI 0.1. 24h post-infection, cells were labelled with Hoescht and with antibodies against dsRNA (detecting viral RNA, in green) and HA (detecting DAXX, in red). When indicated, cells were treated with 50 μ M of GRL0617 at the time of infection. Scale bars correspond to 10 μ m. **d-f: Nsp3 induces DAXX degradation.** **D:** 293T-ACE2 cells were transfected with 1 μ g of the indicated viral proteins. After 24h, the levels of DAXX and GAPDH were analyzed by Western Blot. **E:** 293T-ACE2 cells were transfected with the indicated amounts of Nsp3. After 24h, the levels of DAXX and GAPDH were analyzed by Western Blot. **f:** 293T-ACE2 cells were transfected with 1 μ g of Nsp3. 6 hours post transfection, cells were also, when indicated, treated with 50 μ M of GRL0617. 24h after transfection, the levels of DAXX and GAPDH were analyzed by Western Blot.

We tested the ability of several viral proteins to induce DAXX degradation upon overexpression. We could not use previously described (Gordon *et al.* 2020) lentiviral constructs expressing individual proteins since lentiviral replication is known to be inhibited by DAXX (Dutrieux *et al.* 2015). Moreover, this lentiviral plasmid collection did not include the

viral protease Nsp3, our prime candidate for DAXX degradation. We thus selected proteins based on the availability of expressing plasmids via Addgene.

These novel results show that nsp3 overexpression, but not over-expression of the other viral proteins tested, leads to DAXX degradation, and that this effect is dose-dependent (novel **Fig. 6d-e**). We also show that treatment of cells with GRL0617, an inhibitor of PLpro (here used at two different concentrations), but not with masitinib, the 3CLpro inhibitor, prevents DAXX degradation induced by SARS-CoV-2 infection (**Fig. 6b**) or Nsp3 overexpression (**Fig. 6f**). Treatment with MG132, a well-characterized inhibitor of the proteasome, also prevented DAXX degradation (**Fig. 6b**). Moreover, SARS-CoV-2 infection had no effect on DAXX mRNA levels (novel **Fig. S6**). Taken together, these results suggest that PLpro (rather than 3CLpro or another viral protein) degrades DAXX in a proteasome-dependent mechanism.

Supplementary Figure 6: DAXX mRNA levels are not affected by SARS-CoV-2 replication.

A549-ACE2 WT cells were infected with SARS-CoV-2 at MOI 1. Cellular monolayers were harvested at the indicated time points and total RNA was extracted. The levels of DAXX RNA were determined by qRT-PCR and normalized against GAPDH levels. The mean of 3 independent experiments performed in triplicates is shown. Statistics: 2-way ANOVA using Sidak's test. ns p-value > 0.05.

3. IFN itself can inhibit cell replication and induce cell death. Therefore it is not ideal that the authors compared virus+IFN to no virus/no IFN. The authors should compare virus+IFN to no virus+IFN.

In our experimental setting, we did not observe IFN-induced cell death. As we show in **Fig. S1**, the FSC/SSC profiles of non-infected cells looked similar to those of IFN-treated, SARS-CoV-2 infected cells.

As in our previous CRISPR/Cas9 screens (Roesch *et al.* 2018 ; OhAinle *et al.* 2018), non-infected cells treated with IFN were not included in the screen. We believe that IFN treatment alone should not select for specific knock-outs. Importantly, the results of the screen validate our approach, since we identified expected IFN-signaling genes (IFNAR1, STAT1, STAT2, IRF9) and genes previously known to impact SARS-CoV-2 replication (such as CTSL and LY6E) as our most prominent hits.

4. What is the specificity of DAXX for SARS-CoV-2? Testing the antiviral activity of DAXX against other coronaviruses (or RNA viruses) would provide additional insight into the mechanism and breadth of DAXX function.

We have tested the effect of DAXX on the replication of SARS-CoV and MERS-CoV (novel **Fig. 2f**). We also evaluated the effect of knocking-out DAXX on the replication of a positive-strand RNA virus (Yellow Fever virus, YFV) and a negative-strain RNA virus (Measles virus, MeV) by measuring the number of cells positive for viral protein by flow cytometry analysis (novel **Fig. 2e**). This novel set of experiments revealed that DAXX seems to exhibit some degree of specificity, as it had no effect on YFV, MeV and MERS-CoV, but significantly inhibited SARS-CoV and SARS-CoV-2 replication.

Figure 2: DAXX is a restriction factor for SARS-CoV-2. **e:** A549-ACE2 WT or DAXX KO cells were infected in triplicates with Yellow Fever Virus (YFV, Asibi strain, MOI of 0.3) or with Measles Virus (MeV, Schwarz strain expressing GFP, MOI of 0.2). At 24h p.i., the percentages of cells positive for viral protein E (YFV) or GFP (MeV) was assessed by flow cytometry. The mean of 3 independent experiments is represented. **f:** WT or DAXX KO cells were infected in triplicates at an MOI of 0.1 with SARS-CoV, SARS-CoV-2 or MERS-CoV. Supernatants were harvested at 72h p.i. Supernatants were heat inactivated prior to quantification by qRT-PCR. Serial dilutions of a stock of known infectious titer was used as a standard. The mean of 2 independent experiments is represented. Statistics: 2-way ANOVA using Dunnett's test, * = p-value < 0.05, *** = p-value < 0.001, **** = p-value < 0.0001

Minor revisions

5. *-DAXX is known to regulate apoptosis. It would be helpful to know if cells with altered levels of DAXX show differences in cell death.*

While we have not strictly measured the levels of apoptosis in DAXX KO cells, we did not observe significant cell death in these cells, as shown in the flow cytometry plots below, which were obtained while performing the YFV experiments (see novel **Fig. 2e** above). This is consistent with previously published data (Sudharsan *et al.* 2012, cited in our manuscript), which suggested that DAXX actually potentiates apoptosis.

6. For table 1, was this N=1? What is the variance of editing amongst repeats?

The % of editing was indeed measured by Sanger sequencing only once at the time of the first submission. We have evaluated two more times the % of editing of all cell lines and have added this new data in **Table 1**.

Gene	% of alleles edited
LY6E	96 ± 1,73
DAXX	79,67 ± 2,52
APOL6	99 ± 0
HERC5	97 ± 0
CTSL	91 ± 1
IFI6	88,33 ± 0,58
IFNAR1	76,67 ± 3,21

Table 1: Gene editing efficiency. The frequency of editing was determined using Sanger sequencing and ICE analysis. Values are represented as mean ± SD (n=3).

7. Microscopy images should be quantified in Figure 3.

The quantification of microscopy images of former **Fig.3a** (now **Fig. 4a**) is now shown in **Fig.4b**, and indicates that DAXX overexpression reduces SARS-CoV-2 infection.

Figure 4: DAXX restriction of SARS-CoV-2 is SUMOylation independent. a-c: DAXX overexpression restricts SARS-CoV-2. 293T-ACE2 cells were transfected with DAXX WT. 24h after transfection, cells were infected with the mNeonGreen fluorescent reporter SARS-CoV-2 at the indicated MOI. Cells were either visualized with an EVOS fluorescence microscope (**a-b**) or stained with an HA-antibody detecting DAXX and imaged by confocal microscopy (**c**). Scale bars correspond to 200 μ m (**a**) and 30 μ m (**c**). Images shown in (**a**) were quantified using ImageJ software (**b**). Data show the mean \pm SD of Fluorescence integrated densities. The analysis was performed on around 200 cells from 3 different fields.

8. -In figure 2, the authors assess antiviral activity of “ISGs” but how many of these genes are actually ISGs in the A549s tested? DAXX is not an ISG in A549s so this should be clarified.

Indeed, the sgRNA library we designed (and which is described in more detail in our previous publication, OhAinle *et al.* 2018) includes genes that are only weakly induced by IFN in PBMCs and/or T cells. This was done on purpose, since it is known that the expression of some ISGs can be more or less sensitive to IFN depending on cell types. This information is mentioned in the introduction:

“ This library includes more ISGs than most published libraries, as the inclusion criteria was less stringent than in previous studies (fold-change in gene expression in THP1 cells, primary CD4+ T cells or PBMCs \geq 2). Therefore, some genes present in the library may not be genuine ISGs in A549 cells.”

The specific case of DAXX is also now mentioned:

“ DAXX was originally included in the PIKA library, although its expression is only weakly induced by IFN in some human cell types (32,44) ”

9. The term “PFU equivalents” should be changed to “viral RNA” or “viral genomes” The PFU to RNA ratio can vary wildly (by more than 6-logs) across viral stocks, cell types, etc so this is not appropriate.

The number of genome copies has now been calculated using purified viral RNA, as described in the methods section, and the appropriate figure axes have been re-labelled accordingly.

10. -For reader clarity, it would be preferable to relabel viruses in Figure 2d with more commonly known names i.e. B.1.1.7. Also, “historical strain “would be better as “reference virus” or “prototype virus” (or just give the actual name of the virus).

We have changed the names of all strains used in this study to follow the recently published WHO guidelines, as follows:

- BetaCoV/France/IDF0372/2020 (historical): Lineage B
- hCoV-19/France/IDF-IPP11324/2020 (or UK): Lineage B.1.1.7 (Alpha)
- hCoV-19/France/PDL-IPP01065/2021 (20H/501Y.V2 or SA): Lineage B.1.351 (Beta)
- hCoV-19/Japan/TY7-501/2021 (20J/501Y.V3 or Brazil): Lineage P1 (Gamma)

11. -The label of the Y axis on Figure 2e is confusing. Does a value of 2 mean 100x more viral transcript than GAPDH mRNA or does it mean 100x more viral transcript that the NTC?

This panel has been removed and replaced with the novel **Fig. 3** showing the levels of viral transcripts over time. In this figure, the Y axis corresponds to the fold change of viral RNA normalized against GAPDH (in a log scale). These new results clearly show that an increase in viral transcript levels around 8-10h p.i. in DAXX KO cells, as compared to WT cells.

Figure 3: DAXX restricts SARS-CoV-2 transcription. ACE2 WT or DAXX KO cells were infected at MOI 1 in triplicate. Cell monolayers were harvested at the indicated time points, and total RNA was extracted. The levels of viral RNA (a: 5' UTR; b: RdRp) were determined by RT-qPCR and normalized against GAPDH levels. The mean of 3 independent experiments is represented. Statistics: Dunnett's test on a linear model, * p-value < 0.05, ** p-value < 0.01, *** p-value < 0.001.

12. - In figure 1b, what is the $-\log_{10}$ label on the Y-axis referring to (FDR? P-value?)?

The Y axis of **Fig.1b** represents $-\log_{10}$ of the MAGeCK score, which is a metric commonly represented for analysis of CRISPR screens (for instance, in OhAinle *et al.* 2018) and which is highly correlated to FDR and p-value. We have clarified this by re-labeling the axes “ $-\log_{10}$ (positive MAGeCK score)” and “ $-\log_{10}$ (negative MAGeCK score)”.

13. -In figure 1c, what does the $\text{Log}_2 \text{FC}$ refer to exactly? Guide enrichment of the virus+IFN condition relative to the no virus and no IFN condition? Is it the spike+ vs the spike- cells?

This represents the enriched sgRNAs of the spike-positive cells of the virus+IFN condition relative to the original population of cells (no IFN, no virus). This has been clarified in the figure legend:

“ For the indicated genes, the enrichment ratio of the 8 sgRNAs present in the library was calculated as the MAGeCK normalized read counts in infected cells divided by those in the original pool of cells and is represented in \log_2 fold change. “

14. Line 31: MDA5 needs to be in all caps.

This was corrected.

Reviewer #3 (Remarks to the Author):

Using a wider ISG CRISPR sgRNA library, Kain et al. identified genes, specifically the DAXX, as restriction factors for SARS-CoV-2 infection. DAXX is a known antiviral host factor against DNA viruses, but its function is unrelated to the SUMOylation pathway. In addition, the authors showed that the activity of DAXX on SARS-CoV-2 is IFN-independent, and the protein is degraded upon virus infection. Generally, the finding of DAXX as antiviral host factor for SARS-CoV-2 is novel, but the mechanisms of action remains to be further explored.

We thank the reviewer for his/her kind comments. We have performed additional experiments, detailed below, exploring the mechanism of action of virus-induced DAXX degradation and of DAXX restriction.

One life cycle of coronavirus infection is around 8-12h, thus, the reduced viral transcript measured at 72h post-infection as shown in Fig.2e and probably in Fig. S4c (not sure about the time point) is too late to draw the conclusion that DAXX targets viral transcription (line 148). Although the results indicated that DAXX reduced the virus production, viral transcript, re-located to the cytoplasm and co-stained with the viral dsRNA, more data is required to support the notion that DAXX affects the virus replication or transcription, or probably affects the early entry step. RNA replicon assay, genomic RNA transfection to bypass the entry, or virion binding/internalization assay may be required to demonstrate.

We agree with the reviewer that measuring viral RNA yield at 72h p.i. was not sufficient to claim that DAXX restricts SARS-CoV-2 at the step of viral transcription. This claim was also based on data shown in former Fig. S4c (now Fig. S5) in which we measured viral transcripts at 24h p.i. in DAXX overexpressing cells. To further demonstrate that DAXX acts at the step of viral transcription, we performed the experiment suggested by the reviewer, which is now shown in the new Fig. 3. These novel results clearly show a significant increase in viral transcript levels starting between 8h and 10h p.i. in DAXX KO cells. We did not observe an effect of DAXX on viral replication at earlier time points, arguing that DAXX acts at the step of transcription rather than on viral entry, further validating our previous results.

Figure 3: DAXX restricts SARS-CoV-2 transcription. A549-ACE2 WT or DAXX KO cells were infected at a MOI 1 in triplicate. Cell monolayers were harvested at the indicated time points, and total RNA was extracted. The levels of viral RNA (a: 5' UTR; b: RdRp) were determined by RT-qPCR and normalized against GAPDH levels. The mean of 3 independent experiments is represented. Statistics: Dunnett's test on a linear model, * p-value < 0.05, ** p-value < 0.01, *** p-value < 0.001.

The phenomenon that DAXX could be degraded upon virus infection is not new, as adenovirus and FMDV have similar feature. It is of significance to further figure out how SARS-CoV-2 infection decreases the DAXX protein level or probably the RNA level.

While we agree that degradation of DAXX following a viral infection was described in the past, our work is the first report of this phenomenon for coronaviruses. DAXX degradation by foot-and-mouth disease Virus (FMDV) PLpro was briefly mentioned in the Saiz *et al.* 2021 paper (cited in our manuscript), although the authors did not perform degradation experiments.

In **Fig. 6b**, we now show that treating cells with GRL0617, an inhibitor of PLpro (here used at two different concentrations), but not with masitinib, the 3CLpro inhibitor, prevents SARS-CoV-2-mediated degradation of DAXX. In addition, we have directly tested the effect of several viral proteins on DAXX degradation, and have found that overexpression of Nsp3, which encodes PLpro, leads to DAXX degradation in a dose dependent fashion (**Fig. 6d-e**), a process that was inhibited by GRL0617 (**Fig.6f**). Moreover, we found that treatment with MG132, a well-characterized inhibitor of the proteasome, also prevented DAXX degradation (**Fig. 6b**). We are also showing that SARS-CoV-2 infection had no effect on DAXX mRNA levels (**Fig. S6**). Taken together, these novel results strongly indicate that PLpro (rather than 3CLpro or another viral protein) degrades DAXX in a proteasome-dependent mechanism.

Figure 6: SARS-CoV-2 PLpro induces the proteasomal degradation of DAXX. **a: DAXX degradation after infection.** 293T-ACE2 cells were infected with SARS-CoV-2 at MOI 0.1. After 24h, cells were harvested and levels of DAXX, Lamin B, HSP90, Actin, GAPDH, Tubulin, TRIM22, RIG-I and of the viral protein spike were analyzed by Western Blot. **b: GRL0617 and MG132 treatment restores DAXX expression.** 293T-ACE2 cells were infected with SARS-CoV-2 at MOI 0.1. When indicated, cells were pretreated 2h before infection with GRL0617 (at the indicated concentrations), or with MG132 (10 μM), a proteasome inhibitor, or Masitinib (10 μM) a 3CL inhibitor. After 24h, cells were harvested and levels of DAXX, GAPDH and of the viral protein spike were analyzed by Western Blot. **c: GRL0617 treatment restores DAXX localization.** 293T-ACE2 cells were infected with SARS-CoV-2 at MOI 0.1. 24h post-infection, cells were labelled with Hoescht and with antibodies against dsRNA (detecting viral RNA, in green) and HA (detecting DAXX, in red). When indicated, cells were treated with 50 μM of GRL0617 at the time of infection. Scale bars correspond to 10 μm. **d-f: Nsp3 induces DAXX degradation.** **D:** 293T-ACE2 cells were transfected with 1 μg of the indicated viral proteins. After 24h, the levels of DAXX and GAPDH were

analyzed by Western Blot. **E:** 293T-ACE2 cells were transfected with the indicated amounts of Nsp3. After 24h, the levels of DAXX and GAPDH were analyzed by Western Blot. **f:** 293T-ACE2 cells were transfected with 1 μg of Nsp3. 6 hours post transfection, cells were also, when indicated, treated with 50 μM of GRL0617. 24h after transfection, the levels of DAXX and GAPDH were analyzed by Western Blot.

Supplementary Figure 6: DAXX mRNA levels are not affected by SARS-CoV-2 replication.

A549-ACE2 WT cells were infected with SARS-CoV-2 at a MOI of 1. Cellular monolayers were harvested at the indicated time points and total RNA was extracted. The levels of DAXX RNA were determined by qRT-PCR analysis and normalized against GAPDH levels. The mean of 3 independent experiments performed in triplicates is shown. Statistics: 2-way ANOVA using Sidak's test. ns = p-value > 0.05.

Reviewers' Comments:

Reviewer #1:

Remarks to the Author:

The authors have addressed all points thoroughly and thereby improved the manuscript substantially.

Reviewer #2:

Remarks to the Author:

In the revised manuscript by Mac Kain et al, the authors make substantial progress in understanding the mechanism of DAXX restriction of SARS-CoV-2. They find that DAXX chaperone activity is required and that DAXX acts post-entry. They also show that DAXX chaperone activity and the proteasome are important for function. This is a vastly improved manuscript and the authors should be commended on such a nice paper. I have only minor comments.

1) How many independent experiments were performed for each figure? This should be stated in the figure legends (ie Fig 6 western blots. Were they N=1?)

2) Reviewer 2 comment 3: Since the authors did not assess the effect of IFN in the absence of virus, they should mention this briefly as a limitation in the text. IFN alone can yield IFNAR, STAT1, JAK, etc (see Doench et al Nature Biotech 2016 PMID 267801880).

Reviewer #3:

Remarks to the Author:

One concern about the revised manuscript is the NSP3-mediated DAXX degradation in Figure 6. The western blot should be developed using like fluorescence-based quantification of protein bands in Fig 6d, e and f. In Fig 6d, both pCDNA and nsp3 showed decreased DAXX level when compared to other lanes.

REVIEWER COMMENTS

Reviewer #1 (Remarks to the Author):

The authors have addressed all points thoroughly and thereby improved the manuscript substantially.

We thank reviewer #1 for his/her kind comment.

Reviewer #2 (Remarks to the Author):

In the revised manuscript by Mac Kain et al, the authors make substantial progress in understanding the mechanism of DAXX restriction of SARS-CoV-2. They find that DAXX chaperone activity is required and that DAXX acts post-entry. They also show that DAXX chaperone activity and the proteasome are important for function. This is a vastly improved manuscript and the authors should be commended on such a nice paper. I have only minor comments.

We thank reviewer #2 for his/her kind comment.

1) How many independent experiments were performed for each figure? This should be stated in the figure legends (ie Fig 6 western blots. Were they N=1?)

At least 2 or 3 independent experiments were performed for Western Blot and confocal microscopy analyses. We have updated the figure legends of **Fig. 4**, **Fig. 5** and **Fig. 6** to indicate the number of experiments performed.

Figure 4: DAXX restriction of SARS-CoV-2 is dependent on its chaperone activity but SUMOylation-independent. **a:** Schematic of the DAXX mutants used. The fifteen lysine residues of DAXX 15KR have been mutated to arginine. DAXX Δ SIM lacks the 732-740 C-terminal residues. Both mutants were described in (48). DAXX Δ D/E is lacking its 414-505 domain and has been described in (52) **b-c:** SUMOylation-deficient DAXX mutants are still able to restrict SARS-CoV-2. 293T-ACE2 cells were transfected with HA-DAXX WT; HA-DAXX 15KR; HA-DAXX Δ SIM; or with HA-NBR1 as negative control plasmid. 24h after transfection, cells were

infected with SARS-CoV-2 at an MOI of 0.1. When indicated, cells were treated with remdesivir at the time of infection. After 24h or 48h, infected cells were double-stained recognizing dsRNAs (to read out infection) and HA (to read out transfection efficiency) and acquired by flow cytometry. The percentage of infected cells among HA-positive (transfected) cells for one representative experiment is shown in **b**, for the mean of 3 independent experiments in **c**. Statistics: one-way ANOVA using Dunnett's test, Holm corrected, ns = p-value > 0.05, * = p-value < 0.05, ** = p-value < 0.01, *** = p-value < 0.001. **d-e: The chaperone activity of DAXX is required for SARS-CoV-2 restriction.** 293T-ACE2 cells were transfected with DAXX WT or with the DAXX Δ D/E mutant. 24h after transfection, cells were infected with SARS-CoV-2 mNeonGreen at MOI 1. After 24h, the levels of Spike and GAPDH levels were analyzed by Western Blot (**d**). A Western Blot representative of 3 independent experiments is shown. In parallel, SARS-CoV-2 replication levels were measured by RT-qPCR targeting the 5' UTR and normalized against RPL13A transcript levels (**e**). The mean of 4 independent experiments performed in duplicate is represented. Statistics: 1-way ANOVA using Dunnett's test, ns = p-value > 0.05, **** = p-value < 0.0001.

Figure 5: SARS-CoV-2 infection induces DAXX cytoplasmic re-localization to sites of viral replication. a-c: DAXX overexpression restricts SARS-CoV-2. 293T-ACE2 cells were transfected with DAXX WT. 24h after transfection, cells were infected with the mNeonGreen fluorescent reporter SARS-CoV-2 at the indicated MOI. Cells were either visualized with an EVOS fluorescence microscope (**a-b**) or stained with an HA-antibody detecting DAXX and imaged by confocal microscopy (**c**). Scale bars correspond to 200 μ m (**a**) and 30 μ m (**c**). Images shown in (**a**) were quantified using ImageJ software (**b**). Data shows the mean +/- SD of Fluorescence integrated densities. The analysis was performed on around 200 cells from 3 different fields. Images are representative of 3-6 different fields from 2 independent experiments. **d: Relocalization of endogenous DAXX during SARS-CoV-2 infection.** 293T-ACE2 cells were infected with SARS-CoV-2 at the indicated MOI 1. 24h post-infection, cells were labelled with Hoescht and with antibodies against dsRNA (detecting viral RNA, in green) and HA (detecting DAXX, in red). When indicated, the high-resolution Airyscan mode was used. Scale bars correspond to 10 μ m for confocal images, and 2 μ m for the high-resolution images. Images are representative of 3-6 different fields from 2 independent experiments.

Figure 6: SARS-CoV-2 PLpro induces the proteasomal degradation of DAXX. **a: DAXX degradation after infection.** 293T-ACE2 cells were infected with SARS-CoV-2 at MOI 0.1. After 24h, cells were harvested and levels of DAXX, Lamin B, HSP90, Actin, GAPDH, Tubulin, TRIM22, RIG-I and of the viral protein spike were analyzed by Western Blot. A Western Blot representative of 3 independent experiments is shown. **b: GRL0617 and MG132 treatment restores DAXX expression.** 293T-ACE2 cells were infected with SARS-CoV-2 at MOI 0.1. When indicated, cells were pretreated 2h before infection with GRL0617 (at the indicated concentrations), or with MG132 (10 μ M), a proteasome inhibitor, or Masitinib (10 μ M) a 3CL inhibitor. After 24h, cells were harvested and levels of DAXX, GAPDH and of the viral protein spike were analyzed by Western Blot. A Western Blot representative of 3 independent experiments is shown. **c: GRL0617 treatment restores DAXX localization.** 293T-ACE2 cells were infected with SARS-CoV-2 at MOI 0.1. 24h post-infection, cells were labelled with Hoescht and with antibodies against dsRNA (detecting viral RNA, in green) and HA (detecting DAXX, in red). When indicated, cells were treated with 50 μ M of GRL0617 at the time of infection. Scale bars correspond to 10 μ m. Images are representative of 3-6 different fields from 2 independent experiments. **d-f: Nsp3 induces DAXX degradation.** **d:** 293T-ACE2 cells were transfected with 1 μ g of the indicated viral proteins. After 24h, the levels of DAXX and GAPDH were analyzed by Western Blot. **e:** 293T-ACE2 cells were transfected with the indicated amounts of Nsp3. After 24h, the levels of DAXX and GAPDH were analyzed by Western Blot. **f:** 293T-ACE2 cells were transfected with 1 μ g of Nsp3 or of pcDNA. 6 hours post transfection, cells were also, when indicated, treated with 50 μ M of GRL0617. 24h after transfection, the levels of DAXX and GAPDH were analyzed by Western Blot. Representative Western Blot from 2 independent experiments are shown. The quantification of band intensity for Fig. 6d-f is shown in Fig. S8b, S8d and S8e.

We have also performed quantifications of Western Blot analyses from Fig.6d-f, which are now shown in Fig.S8b, Fig. S8d and Fig. S8e.

Supplementary Figure 8: Expression levels of Nsp-mCherry fusion proteins in 293T-ACE2 cells. **a:** The expression of SARS-CoV-2 Nsp proteins fused to mCherry (Fig. 6d) was evaluated by flow cytometry. **b:** Quantification of Western Blot analyses from Fig. 6d. The intensity of the DAXX and GAPDH bands was determined using the Image Lab Software. The mean \pm SD of the DAXX/GAPDH ratio is represented (n=2). **c:**

The expression of SARS-CoV-2 Nsp3-mCherry (Fig. 6e) was evaluated by flow cytometry. d-e: Quantification of Western Blot analyses from Fig. 6e is shown in (d) and of Fig. 6f in (e). The intensity of the DAXX and GAPDH bands was determined using the Image Lab Software. The mean \pm SD of the DAXX/GAPDH ratio is represented (n=2).

2) Reviewer 2 comment 3: Since the authors did not assess the effect of IFN in the absence of virus, they should mention this briefly as a limitation in the text. IFN alone can yield IFNAR, STAT1, JAK, etc (see Doench et al Nature Biotech 2016 PMID 267801880).

We have added two sentences in the discussion section to mention this potential limitation:

“ One potential caveat to our screen is that it compared IFN-treated infected cells to non-infected untreated cells (rather than IFN-treated non-infected cells). Although we validated this approach in previous studies (34,60), it may cause enrichment of genes induced by IFN, but not antiviral against SARS-CoV-2 per se. “

Reviewer #3 (Remarks to the Author):

One concern about the revised manuscript is the NSP3-mediated DAXX degradation in Figure 6. The western blot should be developed using like fluorescence-based quantification of protein bands in Fig 6d, e and f. In Fig 6d, both pCDNA and nsp3 showed decreased DAXX level when compared to other lanes.

The quantifications of Western Blot analyses from Fig.6d-f are now shown in Fig.S8b, Fig. S8d and Fig. S8e. They were performed on 2 independent experiments using the Image Lab Software (Bio-Rad Laboratories).

Supplementary Figure 8: Expression levels of Nsp-mCherry fusion proteins in 293T-ACE2 cells.

a: The expression of SARS-CoV-2 Nsp proteins fused to mCherry (Fig. 6d) was evaluated by flow cytometry. **b:** Quantification of Western Blot analyses from Fig. 6d. The intensity of the DAXX and GAPDH bands was determined using the Image Lab Software. The mean \pm SD of the DAXX/GAPDH ratio is represented (n=2). **c:** The expression of SARS-CoV-2 Nsp3-mCherry (Fig. 6e) was evaluated by flow cytometry. **d-e:** Quantification of Western Blot analyses from Fig. 6e is shown in (d) and of Fig. 6f in (e). The intensity of the DAXX and GAPDH bands was determined using the Image Lab Software. The mean \pm SD of the DAXX/GAPDH ratio is represented (n=2).